# Mesenchymal stem cell therapy induces FLT3L and CD1c+ dendritic cells in systemic lupus erythematosus patients

Xinran Yuan[1,2,5], Xiaodong Qin[3,5], Dandan Wang[1,5], Zhuoya Zhang[1], Xiaojun Tang[1], Xiang Gao[2], Wanjun Chen[4] & Lingyun Sun[1,2]

Allogeneic mesenchymal stem cells (MSCs) exhibit immunoregulatory function in human autoimmune diseases such as systemic lupus erythematosus (SLE), but the underlying mechanisms remain incompletely understood. Here we show that the number of peripheral tolerogenic CD1c+ dendritic cells (DCs) and the levels of serum FLT3L are significantly decreased in SLE patients especially with lupus nephritis, compared to healthy controls. Transplantation of allogeneic umbilical cord-derived MSCs (UC-MSCs) significantly up-regulates peripheral blood CD1c+DCs and serum FLT3L. Mechanistically, UC-MSCs express FLT3L that binds to FLT3 on CD1c+DCs to promote the proliferation and inhibit the apoptosis of tolerogenic CD1c+DCs. Conversely, reduction of FLT3L with small interfering RNA in MSCs abolishes the up-regulation of tolerogenic CD1c+DCs in lupus patients treated with MSCs. Interferon-γ induces FLT3L expression in UC-MSCs through JAK/STAT signaling pathway. Thus, allogeneic MSCs might suppress inflammation in lupus through up-regulating tolerogenic DCs.

[1] Department of Rheumatology and Immunology, The Affiliated Drum Tower Hospital of Nanjing University Medical School, Nanjing, Jiangsu 210008, PR China. [2] State Key Laboratory of Pharmaceutical Biotechnology, Nanjing University, Nanjing, Jiangsu 210061, PR China. [3] Department of Orthopedics, The Affiliated Drum Tower Hospital of Nanjing University Medical School, Nanjing, Jiangsu 210008, PR China. [4] Mucosal Immunology Section, NIDCR, NIH, Bethesda, MD 20892, USA. [5]These authors contribute equally: Xinran Yuan, Xiaodong Qin, Dandan Wang. Correspondence and requests for materials should be addressed to L.S. (email: lingyunsun@nju.edu.cn)

Systemic lupus erythematosus (SLE) is a systemic auto-immune disease that is characterized by autoantibodies with immune complexes deposition leading to multi-organ injury[1]. In spite of the significant variation of disease severity among SLE patients, the dysregulation of innate and adaptive immune responses are universal[2]. An insight into the underlying pathology is important to develop optimal therapies.

Dendritic cells (DCs) are innate immune cells that play key roles in initiating and regulating immunity by sensing and integrating signals from a wide range of pathogens and tissue damage[3]. Previous studies about DCs in SLE mainly focused on the plasmacytoid DCs (pDCs)[4], but the contribution of conventional DCs (cDCs) to SLE pathogenesis remains largely unknown. Plasticity of DCs with different maturity status and function allows them to be exploited as tolerogenic DCs for their clinical application in autoimmune disease[5]. Previous studies have suggested tolerogenic DC subsets in human displaying immunoregulatory functions, such as CD1c$^+$DCs in human peripheral blood and in lymphoid and non-lymphoid tissues[6]. Compared with other conventional DC subsets, poly I:C-activated CD1c$^+$DCs express low levels of co-stimulatory molecules CD40, CD80, CD83, CD86, as well as stimulatory cytokines such as interleukin (IL)-12, to induce IFN-γ-secreting T cells[7]. In response to Escherichia coli (E.coli) stimulation, human peripheral CD1c$^+$DCs produced only low levels of tumor necrosis factor (TNF), IL-6, IL-12. Instead, they produced high levels of the anti-inflammatory cytokine IL-10 and regulatory molecules indoleamine 2,3-dioxygenase (IDO). These CD1c$^+$DCs suppressed T-cell proliferation in an IL-10-dependent manner[8,9]. Similarly, human liver resident CD1c$^+$DCs produced large amounts of IL-10 upon LPS stimulation and induced immunoregulatory T regulatory cells and IL-4-producing Th2 cells via an IL-10-dependent mechanism[10]. Moreover, human pulmonary CD1c$^+$DCs also exert tolerogenic properties in inducing Treg by IL-27/IL-10/inducible costimulator ligand[11]. Thus, certain CD1c$^+$DCs might be a crucial DC subtype for ameliorating immune dysfunction and maintaining immune homeostasis. Although previous studies have reported the abnormalities of DCs in SLE, the number and function of tolerogenic CD1c$^+$DC subset in lupus patients remain largely unknown.

The cytokine Fms-related tyrosine kinase 3-ligand (FLT3L) is a key regulator of DC commitment in hematopoiesis, which regulates the proliferation, differentiation and apoptosis of haematopoietic cells through the binding to FLT3[12–14]. Not only do cDCs proliferation depend on FLT3L, but also CD1c$^+$DCs, as well as their precursors could be specifically elevated by injection of FLT3L in humans and mice[15,16]. However, researches regarding FLT3L focused mainly on hematopoietic cells and hematological diseases, but few in autoimmune diseases.

Mesenchymal stem cells (MSCs) are non-hematopoietic stem cells that can support the function of hematopoietic stem cells (HSCs) in bone marrow. MSCs possess immunomodulatory properties and functions to immune cells[17], including T, B, NK cells and macrophages[18–20]. MSCs also regulate DC function by inhibiting their co-stimulatory molecule expression and pro-inflammatory cytokine production, leading to the suppression of their capacity to initiate naive T cells and immune responses[21–24]. Moreover, it is reported that the immunosuppressive function of MSCs is enhanced by pro-inflammatory cytokines such as interferon-γ (IFN-γ)[25]. We have previously shown that CD8$^+$T cells derived IFN-γ in SLE patients induced MSCs to secret large amount of IDO that suppressed T cell proliferation[26]. Meanwhile, the serum levels of IFN-γ in SLE patients could predict MSCs therapeutic effect[27]. However, whether MSCs regulate tolerogenic DCs in lupus proinflammatory environment remains unknown.

Here, we determine that the decreased number of tolerogenic CD1c$^+$DCs due to the deficiency of FLT3L contributes to the pathogenesis of SLE. Allogeneic UC-MSCs promote the proliferation and inhibit the apoptosis of tolerogenic CD1c$^+$DCs by up-regulating FLT3L in SLE patients, which could be enhanced by the high level of IFN-γ in lupus environment. Thus, we uncover a previous unrecognized IFN-γ/FLT3L-FLT3/CD1c$^+$DCs axis that mediates the therapeutic benefit of allogeneic UC-MSCs in lupus.

## Results

**CD1c$^+$DCs decrease in SLE patients.** We first investigated the numbers of peripheral blood CD1c$^+$DCs in patients with SLE by flow cytometry. CD1c$^+$DCs were defined as Lin (CD3/19/56/14)$^-$HLA-DR$^+$CD11c$^+$CD1c$^+$ (Fig. 1a). The number of CD1c$^+$DCs in SLE patients was markedly decreased compared to healthy controls (HC) (Fig. 1b, c, d) and patients with rheumatoid arthritis (RA) (Supplementary Fig. 1). The number of CD1c$^+$DCs was negatively correlated with systemic lupus erythematosus disease activity index (SLEDAI) score (Fig. 1e). Additionally, SLE patients with lupus nephritis (LN) showed a further reduced percentage of CD1c$^+$DCs than those without nephritis (Fig. 1f). In addition, the number of CD1c$^+$DCs was negatively correlated to renal indexes namely 24 h proteinuria (Fig. 1g), blood urea nitrogen (BUN) (Fig. 1h) and serum creatinine (Scr) (Fig. 1i). These results all together indicated that CD1c$^+$DCs might participate in the regulation of SLE pathogenesis.

**Decreased FLT3L results in CD1c$^+$DCs deficiency.** Next, we studied the underlying mechanisms for the deficiency of CD1c$^+$DCs in SLE patients. Previous studies have found that FLT3L injection induced expansion of CD1c$^+$DCs in human blood[15], so we determined FLT3L levels in SLE and HC by real-time PCR and ELISA. The mRNA level of FLT3L was significantly decreased in SLE peripheral blood mononuclear cells (PBMCs) (Fig. 2a), and the protein level in SLE serum was also decreased (Fig. 2b). In contrast, the mRNA level of FLT3 (CD135) in PBMCs was significantly increased in SLE patients compared with healthy controls (Fig. 2c). There was a trend towards that FLT3L was negatively correlated with FLT3 (Fig. 2d). Although both FLT3L and CD1c$^+$DCs were significantly decreased, and FLT3 was significantly increased in SLE patients, the changes of FLT3 and its ligand were smaller than that of CD1c$^+$DCs in SLE patients. To determine whether FLT3L could up-regulate CD1c$^+$DCs in vitro, SLE PBMCs were stimulated with different concentrations of recombinant human FLT3L (1, 10, 100 and 500 ng/ml). After stimulation for 48 h, we found that FLT3L up-regulated the number of CD1c$^+$DCs in SLE PBMCs, with statistical difference at a concentration of 10 ng/ml (Fig. 2e, Supplementary Fig. 2). FLT3L could also up-regulate the number of CD1c$^+$DCs in PBMCs obtained from healthy subjects (Supplementary Fig. 3). These data suggested that decreased level of FLT3L in SLE patients might result in the deficiency of CD1c$^+$DCs.

**UC-MSCs ameliorate CD1c$^+$DCs deficiency in SLE patients.** Twenty-one SLE patients, who were refractory to conventional therapies, underwent UC-MSCs transplantation (U-MSCT). The clinical features and medication history of the SLE patients underwent U-MSCT are listed in Tables 1, and 2 and Supplementary Table 1. No serious adverse events were found during 24 h and/or 72 h after infusions of UC-MSCs in any of the 21 SLE patients. No treatment-related mortality or other adverse events occurred during or after U-MSCT, and U-MSCT was well tolerated by all patients.

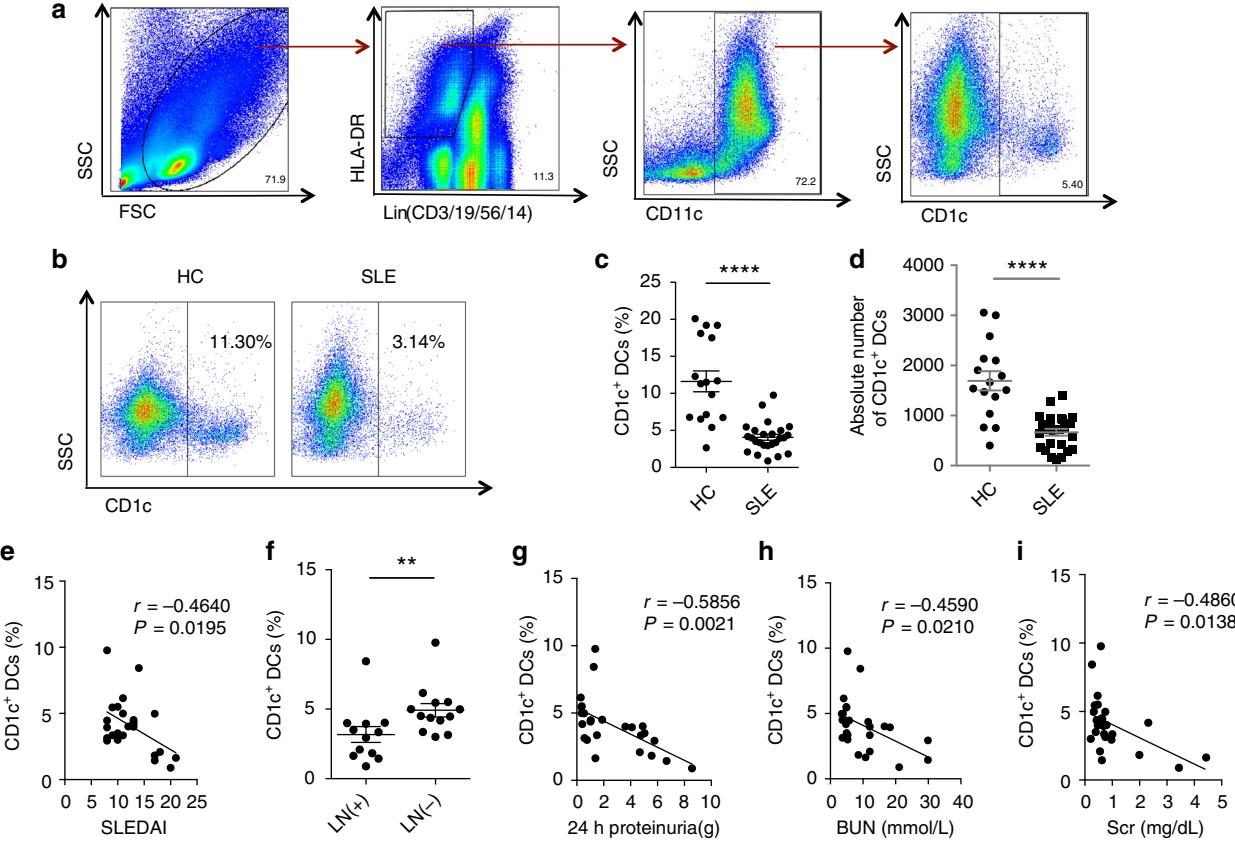

**Fig. 1** CD1c+DCs decrease in patients with SLE. **a** FACS gating strategy used to identify CD1c+DCs, which were defined as Lin(CD3/19/56/14)−HLA-DR+CD11c+CD1c+. **b** Flow cytometry analyses of CD1c+DCs in PBMCs between SLE patients ($n = 25$) and healthy controls (HC) ($n = 16$). **c, d** Quantification of CD1c+DCs in (b). **e** The correlation between the frequency of CD1c+DCs and systemic lupus erythematosus disease activity index (SLEDAI) score was analyzed by Pearson's correlation coefficient ($n = 25$). **f** The percentage of CD1c+DCs was compared between SLE patients with lupus nephritis (LN) ($n = 12$) and without ($n = 13$). **g** The correlation between the frequency of CD1c+DCs and the level of 24 h proteinuria was analyzed by Pearson's correlation coefficient ($n = 25$). **h** The correlation between the frequency of CD1c+DCs and the level of blood urea nitrogen (BUN) was analyzed by Pearson's correlation coefficient ($n = 25$). **i** The correlation between the frequency of CD1c+DCs and the level of serum creatinine (Scr) was analyzed by Pearson's correlation coefficient ($n = 25$). The experiments were repeated ≥3 times (**$p < 0.01$, ****$p < 0.0001$, by $t$-test), SEM. Source data were provided as a Source Data file

The peripheral blood Lin−HLA-DR+CD11c+CD1c+DCs were analyzed before and after U-MSCT. The frequency and absolute number of CD1c+DCs were significantly increased 24 h and 72 h after U-MSCT (Fig. 3a–c, e–g, Supplementary Fig. 4). In addition, serum FLT3L levels also increased after U-MSCT (Fig. 3d, h). The first 11 patients have completed 6-month follow-up. The SLEDAI score remarkably declined after U-MSCT at 6-month follow-up (Fig. 3i). Two patients (patient 2 and 11) showed complete remission with SLEDAI scores <3 at 6-month follow-up. Seven patients showed partial remission with SLEDAI scores <8 at 6-month follow-up. Two patients (patient 7 and 8) had a flare of disease with SLEDAI scores ≥8 at their 6-month follow-up, respectively, among which, one patient had renal transplantation 15 months after MSCs transplantation (patient 7), the other had renal dialysis 12 months after MSCs transplantation (patient 8). The last 10 patients have completed 1-month follow-up. The SLEDAI score remarkably declined after U-MSCT at 1-month follow-up (Supplementary Fig. 5). Six patients showed partial remission with SLEDAI scores <8 at 1-month follow-up (remission group), and the other four patients showed no remission with SLEDAI scores ≥8 at their 1-month follow-up (no remission group). The increase of CD1c+DCs was significantly larger in remission group compared with no remission group (Supplementary Fig. 6). In addition, there were no significant changes of pDCs (Lin−HLA-DR+CD11c−CD123+) or CD141+DCs (Lin−HLA-DR+CD11c+CD141+) after U-MSCT (Supplementary Fig. 7, 8). Furthermore, rather than U-MSCT, the lupus nephritis patients who achieved partial remission after regular medication therapy showed no significant change of CD1c+DCs number (Supplementary Fig. 9).

Since LN patients showed a much lower number of CD1c+ DCs, and the frequency of CD1c+DCs was correlated to renal indexes, we followed-up the renal involvement 6 months after U-MSCT. The level of 24 h proteinuria significantly decreased (Fig. 3j), but the levels of BUN (Fig. 3k) and Scr (Fig. 3l) had no significant change.

Next, PBMCs from patients with SLE were co-cultured with UC-MSCs at ratios of 1:1 and 10:1 in vitro, and we found UC-MSCs significantly up-regulated the frequency of CD1c+DCs regardless of the ratios (Supplementary Fig. 10). Then we co-cultured SLE PBMCs and UC-MSCs at a ratio of 10:1 for 24 h, 48 h, and 72 h, and found UC-MSCs significantly up-regulated the frequency of CD1c+DCs in a time-dependent manner (Fig. 3m). By 72 h, UC-MSCs significantly up-regulated the number of CD1c+DCs when co-cultured with SLE PBMCs (Fig. 3n, Supplementary Fig. 11), along with the increased FLT3L protein in the supernatants (Fig. 3o). CD1c+DCs were up-regulated more apparently in direct cell–cell contact group than in indirect transwell culture group (Fig. 3p). In addition, there

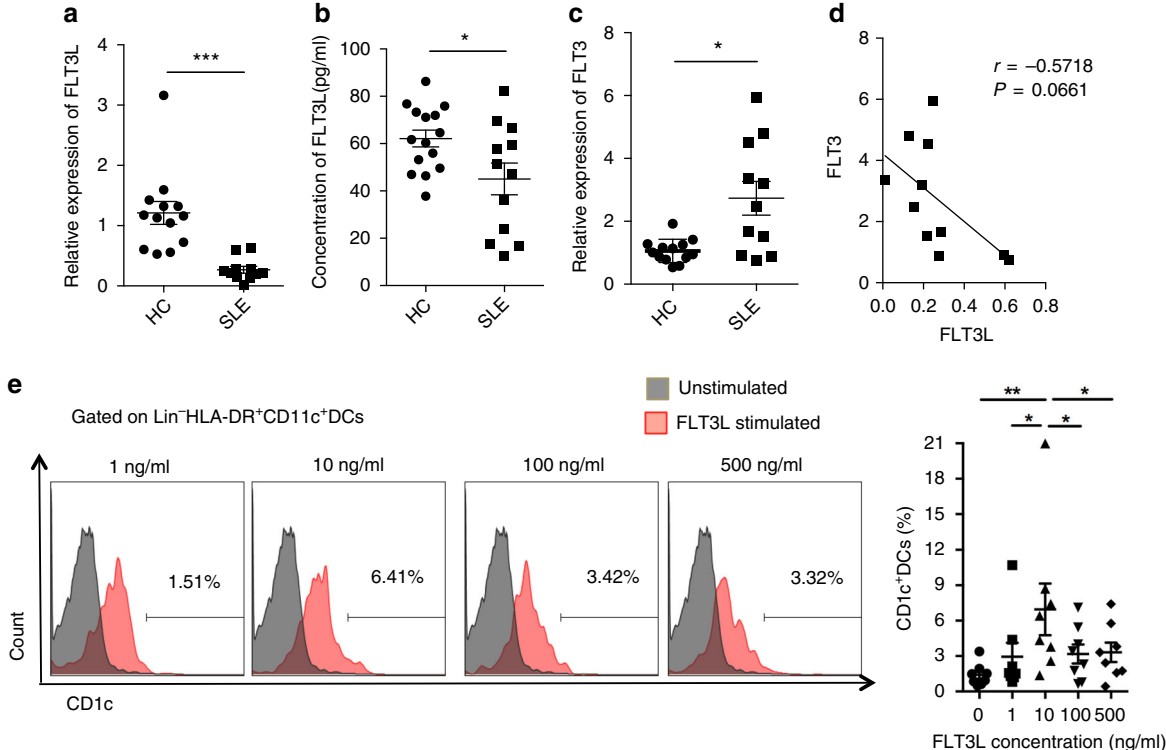

**Fig. 2** FLT3L level is decreased in lupus patients. **a** The mRNA levels of FLT3L in PBMCs from lupus patients ($n = 11$) and HC ($n = 13$) were analyzed by real-time PCR. **b** The protein levels of FLT3L in serum from lupus patients ($n = 12$) and HC ($n = 15$) were analyzed by ELISA. **c** The mRNA levels of FLT3 in lupus ($n = 11$) and HC ($n = 13$) PBMCs were analyzed by real-time PCR. **d** The correlation between the FLT3L and FLT3 in lupus PBMCs was analyzed by Pearson's correlation coefficient ($n = 11$). **e** The PBMCs from SLE patients were stimulated with recombinant human FLT3L with 0 ng/ml, 1 ng/ml, 10 ng/ml, 100 ng/ml, 500 ng/ml, respectively, and after 48 h the percentage of Lin⁻HLA-DR⁺CD11c⁺CD1c⁺DCs in SLE PBMCs was analyzed by flow cytometry ($n = 8$). The experiments were repeated ≥3 times (*$p < 0.05$, **$p < 0.01$, ***$p < 0.001$, by $t$-test or one-way ANOVA), SEM. Source data were provided as a Source Data file

**Table 1 Patients' baseline clinical characteristics**

| Patient | Age/sex | Disease duration (months) | Baseline SLEDAI | Clinical manifestation |
|---|---|---|---|---|
| 1 | 41/F | 120 | 15 | LN, A, C, F, ANA+, anti-dsDNA+, anti-SM+, |
| 2 | 37/F | 36 | 13 | LN, A, C, H, ANA+, anti-dsDNA+ |
| 3 | 26/F | 84 | 14 | LN, A, C, F, H, P, ANA+, anti-SM+, |
| 4 | 26/F | 192 | 15 | LN, C, H, ANA+, anti-dsDNA+, anti-SM+, |
| 5 | 31/M | 48 | 21 | LN, C, H, P, ANA+, anti-dsDNA+, anti-SM+, |
| 6 | 55/F | 180 | 11 | LN, A, C, H, ANA+ |
| 7 | 21/M | 96 | 14 | LN, A, C, F, ANA+, anti-SM+, |
| 8 | 38/F | 84 | 28 | LN, A, C, H, P, V, ANA+, anti-dsDNA+, anti-SM+ |
| 9 | 44/F | 96 | 13 | LN, A, F, H, ANA+, anti-dsDNA+, anti-SM+ |
| 10 | 29/M | 102 | 11 | LN, C, F, ANA+, anti-dsDNA+, anti-SM+ |
| 11 | 26/F | 120 | 13 | LN, C, F, H, P, ANA+, anti-dsDNA+, anti-SM+ |

*A* arthralgia, *ANA* antinuclear antibody, *anti-dsDNA* anti double strand DNA antibody, *C* cytopenia, *F* febrile, *H* hypocomplementemia, *LN* lupus nephritis, *P* polyserositis, *V* vasculitis

were no significant changes of pDCs or CD141⁺DCs in PBMCs when co-cultured with UC-MSCs (Supplementary Figs 12, 13).

In addition, since in vivo the function of CD1c⁺DCs is influenced by their environment and they can be matured into immunogenic cells by other stimuli such as CD40 ligand (CD40L) and toll-like receptor (TLR) activation, both of which are likely to occur in lupus patients, we next investigated the function of the UC-MSCs induced CD1c⁺DCs in the lupus patients, by their functional markers (CD40/CD80/CD83/CD86 expression), ability to present antigen (HLA-DR expression) and cytokines production including IL-10 and TNFα. The peripheral blood

CD1c⁺DCs were analyzed before and after U-MSCT in the last 10 patients (Supplementary Table 1, patient 12–21). We found that after U-MSCT, the expression of the co-stimulatory molecules and HLA-DR of CD1c⁺DCs in freshly isolated peripheral blood of treated SLE did not change significantly. However, LPS-mediated upregulation of CD80, CD86 and HLA-DR expressions of CD1c⁺DCs were significantly decreased in SLE patients after MSCs transplantation compared to before treatment (Fig. 3q, r); similarly, LPS-induced TNFα produced by CD1c⁺DCs in SLE PBMCs were also significantly decreased after MSCs transplantation (Fig. 3s), whereas the level of IL-10 showed

**Table 2 Patients' treatment before and after U-MSCT**

| Patient | Treatment before U-MSCT | Treatment 6-mo after U-MSCT |
|---|---|---|
| 1 | Pred 15 mg/d, CYC 0.8 gm/mo, LEF 20 mg/d, HCQ 0.4 gm/d (×26 mo) | Pred 5 mg/d, HCQ 0.4 gm/d |
| 2 | Pred 30 mg/d, CYC 0.8 gm/mo (×16 mo), MMF 1 gm/d (×14 mo) | Pred 5 mg/d, MMF 1 gm/d, HCQ 0.2 gm/d |
| 3 | Pred 20 mg/d, LEF 20 mg/d, Tacrolimus 3 mg/d, HCQ 0.2 gm/d (×34 mo) | Pred 5 mg/d, Tacrolimus 1.5 mg/d, HCQ 0.2 gm/d |
| 4 | Pred 20 mg/d, CYC 0.8 gm/mo, LEF 20 mg/d, HCQ 0.2 gm/d (×8 mo) | Pred 5 mg/d, MMF 1 gm/d, HCQ 0.2 gm/d |
| 5 | Pred 35 mg/d, MMF 1.5 gm/d, HCQ 0.4 gm/d (×16 mo) | Pred 15 mg/d, HCQ 0.2 gm/d |
| 6 | Pred 10 mg/d, CYC 0.8 gm/mo, HCQ 0.2 gm/d (×12 mo) | Pred 5 mg/d |
| 7 | Pred 15 mg/d, CYC 0.8 gm/mo, MMF 1 gm/d (×8 mo) | Pred 5 mg/d, Cyclosporine 150 mg/d, HCQ 0.2 gm/d |
| 8 | Pred 40 mg/d (×1 mo), then tapered to Pred 30 mg/d, CYC 0.8 gm/mo, LEF 20 mg/d, HCQ 0.2 gm/d (×6 mo) | Pred 10 mg/d, Tacrolimus 2 mg/d, HCQ 0.2 gm/d |
| 9 | Pred 20 mg/d, MMF 1.5 gm/d, HCQ 0.4 gm/d (×17 mo) | Pred 5 mg/d, HCQ 0.2 gm/d |
| 10 | Pred 30 mg/d, CYC 1 gm/mo (×12mo), LEF 20 mg/d (×6 mo) | Pred 10 mg/d, CYC 0.6 gm/2 mo, HCQ 0.2 gm/d |
| 11 | Pred 30 mg/d, Tacrolimus 3 mg/d, HCQ 0.2 gm/d (×12 mo) | Pred 5 mg/d, HCQ 0.4 gm/d |

*CYC* cyclophosphamide, *d* day, *HCQ* hydroxychloroquine, *LEF* leflunomide, *MMF* mycophenolate mofetil, *mo* month, *Pred* prednisone, *U-MSCT* UC-MSCs transplantation

no significant change after U-MSCT (Fig. 3t). Upon CD40L stimulation, the level of TNFα of CD1c+DCs significantly decreased after U-MSCT (Supplementary Fig. 14), while the level of IL-10 showed no significant change after U-MSCT (Supplementary Fig. 15). In addition, we studied the functional change of CD1c+DCs in SLE PBMCs after co-cultured with MSCs. Consistent to the above in vivo data, upon LPS stimulation, the expressions of CD83, CD86, HLA-DR and TNFα production in CD1c+DCs were significantly decreased when co-cultured with UC-MSCs than without UC-MSCs (Fig. 3u, v, w). Whereas there was no significant difference of IL-10 production between the two groups (Fig. 3x).

Collectively, our data demonstrated that in lupus microenvironment, UC-MSCs not only increased the number of CD1c+ DCs, but also drove CD1c+DCs towards a more tolerogenic phenotype.

**UC-MSCs up-regulate CD1c+DCs via FLT3L.** Next, we wanted to verify whether FLT3L played a role in the up-regulation of CD1c+DCs by MSCs. First, we demonstrated UC-MSCs expressed FLT3L by immunofluorescence and flow cytometry (Fig. 4a, b). As expected, by flow cytometry we demonstrated CD1c+DCs expressed FLT3 (Fig. 4c). Moreover, by real-time PCR, compared with CD4+T cells and CD19+B cells, CD1c+DCs expressed much higher level of FLT3 (Fig. 4d). To confirm the role of FLT3L on UC-MSCs in the up-regulation of CD1c+DCs, we inhibited FLT3L expression on UC-MSCs with FLT3L siRNA (Fig. 4e, Supplementary Fig. 16) and found that the reduction of FLT3L abrogated the up-regulation of CD1c+DCs by UC-MSCs (Fig. 4f). Although the expression levels of IL-6 and HGF on UC-MSCs were significantly higher than FLT3L (Supplementary Fig. 17), inhibiting IL-6 or HGF on UC-MSCs with siRNA had no effect on the ability of UC-MSCs to up-regulate CD1c+DCs (Supplementary Figs. 18, 19, 20). These data suggested that FLT3L produced by UC-MSCs predominantly mediated the up-regulation of CD1c+DCs in lupus patients.

To understand the mechanisms by which allogeneic UC-MSCs increase CD1c+DCs, we first exclude the possibility that UC-MSCs indirectly regulate CD1c+DCs via other cells in PBMCs. The UC-MSCs could significantly up-regulate CD1c+DCs in CD4−CD8−PBMCs (T cell-depleted PBMCs), which basically concluded that UC-MSCs up-regulating CD1c+DCs was T cells-independent (Supplementary Fig. 21). Although the expression of FLT3L on PBMCs was significantly higher than that on UC-MSCs (Fig. 4g), the autologous PBMCs (CD1c+DCs depleted) could not up-regulate CD1c+DCs in PBMCs like UC-MSCs

did (Fig. 4h). Hence, we could basically exclude the possibility that UC-MSCs indirectly regulate CD1c+DCs via PBMCs. Then in order to detect whether UC-MSCs directly act on CD1c+DCs, we isolated CD1c+DCs from healthy subjects and cultured with UC-MSCs. We found that UC-MSCs significantly increased the proliferation (Fig. 4i) and decreased the apoptosis of CD1c+DCs (Fig. 4j). Importantly, knock down of FLT3L with specific siRNA significantly reduced the effects of UC-MSCs on the proliferation and apoptosis of CD1c+DCs (Fig. 4k, l). Moreover, we co-cultured UC-MSCs with the PBMCs in which the CD1c+DCs were depleted, and found UC-MSCs could not drive CD1c− PBMCs to differentiate into CD1c+DCs (Supplementary Fig. 22). Our findings suggested a key role of FLT3L in MSCs-mediated increase in tolerogenic CD1c+DCs.

**IFN-γ increases FLT3L in UC-MSCs.** To investigate whether inflammatory microenvironment in lupus patients influence FLT3L expression of UC-MSCs, we stimulated UC-MSCs with IFN-γ (20 ng/ml), as SLE patients show increased IFN-γ[27]. Indeed, IFN-γ enhanced FLT3L expression on UC-MSCs (Fig. 5a). Interestingly, SLE PBMCs and serum also increased the expression of FLT3L on UC-MSCs, while HC PBMCs and serum did not (Fig. 5b, c). The addition of anti-IFN-γ antibody inhibited FLT3L expression on UC-MSCs and consequentially abolished the increase of CD1c+DCs when co-cultured with SLE PBMCs (Fig. 5d, e). Furthermore, stimulation of UC-MSCs with SLE PBMCs activated JAK/STAT signaling pathways and up-regulated FLT3L protein, which was abrogated by anti-IFN-γ antibody (Fig. 5f, g, Supplementary Fig. 23, 24). In addition, AG490, a specific inhibitor of JAK/STAT, decreased the expression of FLT3L in UC-MSCs stimulated with IFN-γ or SLE PBMCs (Fig. 5h). The data altogether indicate that JAK/STAT pathway is involved in the up-regulation of FLT3L on UC-MSCs induced by IFN-γ or SLE PBMCs.

## Discussion

The mechanisms that account for UC-MSCs-mediated immunoregulation in lupus patients remain incompletely understood. Herein we have uncovered that UC-MSCs increased tolerogenic CD1c+ DCs through FLT3L–FLT3 interaction, which could be enhanced by IFN-γ.

In SLE, the breakdown of immune tolerance is widespread and involves autoantigens present in several cell types, making the design of antigen-specific therapy difficult. Current lupus therapies inhibiting both T and B lymphocytes, carry risk of adverse events such as infection[28,29]. B cell-selective monoclonal antibodies targeting CD20

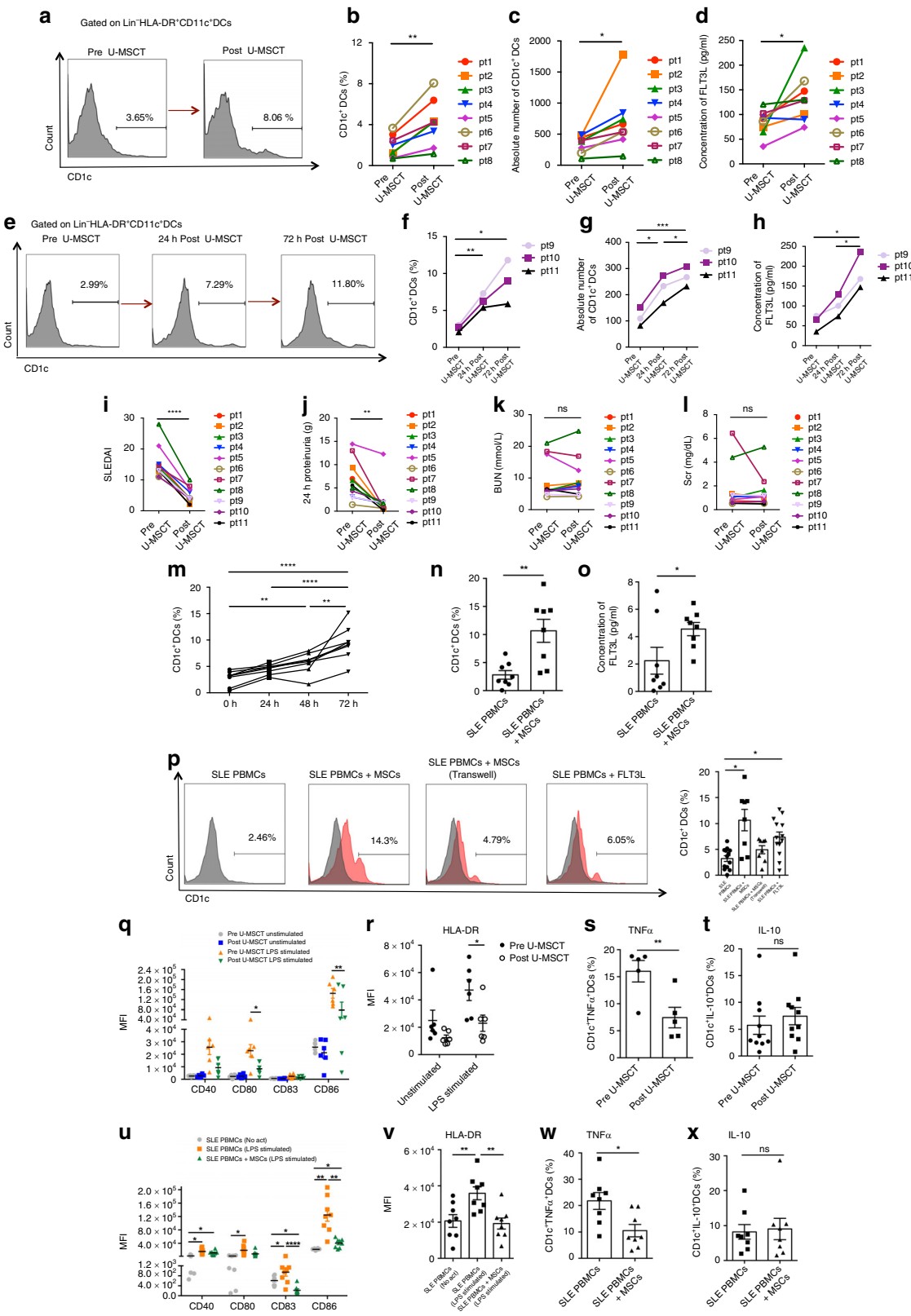

(rituximab) and BAFF (belimumab), despite clinical efficacy in patients with refractory disease, are also associated with long-term treatment-related adverse events[30,31]. All available medications for managing lupus, including belimumab, work through the same mode of action, namely nonspecific suppression of the immune system. Our findings revealed that the number of CD1c+DCs in PBMCs in SLE patients decreased compared with healthy controls and RA patients (Supplementary Fig. 1), and the number of CD1c+DCs was negatively correlated with the SLEDAI score and LN-related indexes (Fig. 1e, f, g, h, i), indicating the crucial role of this tolerogenic DC subset in the pathogenesis of SLE, especially in patients with renal abnormalities.

**Fig. 3** UC-MSCs ameliorate CD1c+DCs deficiency in human SLE. **a–h** Eleven refractory lupus patients were given UC-MSCs transplantation (U-MSCT). Quantification of Lin−HLA-DR+CD11c+CD1c+DCs by flow cytometry (**b**, **c**, **f**, **g**), and serum FLT3L level by ELISA (**d**, **h**) at 24 h ($n = 8$) and 72 h ($n = 3$) after U-MSCT. **i–l** The SLEDAI score (**i**), level of 24 h proteinuria (**j**), BUN (**k**) and Scr (**l**) of total 11 SLE patients were evaluated before and 6 months after U-MSCT ($n = 11$). **m** SLE PBMCs were co-cultured with UC-MSCs for 24 h, 48 h and 72 h, and the frequency of CD1c+DCs in PBMCs was analyzed by flow cytometry at different time points ($n = 8$). **n**, **o** After co-cultured with UC-MSCs for 72 h, the number of CD1c+DCs in SLE PBMCs was analyzed by flow cytometry (**n**), and the level of FLT3L in the culture supernatant was analyzed by ELISA (**o**) ($n = 8$). **p** SLE PBMCs were cultured alone, directly or indirectly (transwell) co-cultured with UC-MSCs, or stimulated with FLT3L (10 ng/ml). After 72 h the percentage of CD1c+DCs in SLE PBMCs was analyzed by flow cytometry ($n = 13$ for SLE PBMCs and SLE PBMCs + FLT3L group; $n = 8$ for SLE PBMCs + MSCs and SLE PBMCs + MSCs (transwell) group). **q**, **r** The mean fluorescence intensity (MFI) of CD40, CD80, CD83, CD86 (**q**) and HLA-DR (**r**) on Lin−HLA-DR+ CD11c+ CD1c+DCs were analyzed before and 24 h after U-MSCT with or without LPS stimulation ($n = 6$). **s**, **t** The production of TNFα ($n = 5$) (**s**) and IL-10 ($n = 10$) (**t**) by Lin−HLA-DR+ CD11c+ CD1c+DCs before and 24 h after U-MSCT. **u**, **v** The MFI of CD40, CD80, CD83, CD86 (**u**) and HLA-DR (**v**) on Lin−HLA-DR+CD11c+CD1c+DCs were analyzed before and 72 h after co-cultured with UC-MSCs ($n = 8$). **w**, **x** The production of TNFα (**w**) and IL-10 (**x**) by Lin−HLA-DR+CD11c+CD1c+DCs before and 72 h after co-cultured with UC-MSCs ($n = 8$). The experiments were repeated ≥3 times (*$p < 0.05$, **$p < 0.01$, ***$p < 0.001$, ****$p < 0.0001$, ns, not significant by t-test or one-way ANOVA), SEM. Source data were provided as a Source Data file

As a key regulator of DCs hematopoiesis, FLT3L regulates the differentiation, proliferation and apoptosis of peripheral DCs in human and mouse[32,33]. In this study, we found that the level of FLT3L significantly decreased in peripheral blood in patients with SLE, and human recombinant FLT3L stimulation significantly increased the proportion of CD1c+DCs in lupus PBMCs in vitro. Thus we considered that the defect of FLT3L was at least partly contributed to the deficiency of CD1c+DCs in lupus patients. At odds with our findings, Nakamura and his colleagues reported the serum FLT3L level was slightly higher in patients with SLE compared with normal subjects, but the difference was not statistically significant[34]. This discrepancy could be due to differences in cohort characteristics, the experimental protocols used, or both.

Allogeneic UC-MSCs show unique immunoregulatory functions by suppressing T cell proliferation and inducing the generation of Treg, inhibiting B cell function and IgG production, or promoting the phagocytic activity of macrophage[35–37]. However, whether DCs could be regulated by UC-MSCs in lupus microenvironment is rarely known. Our findings revealed that UC-MSCs significantly up-regulated the number of CD1c+DCs via FLT3L in a time-dependent manner (Fig. 3f, g, m). In support of this conclusion, we demonstrated that UC-MSCs transplantation remarkably up-regulated the level of serum FLT3L along with an increase of CD1c+DCs in lupus patients. Meanwhile, siRNA against FLT3L could abrogate the up-regulation of CD1c+DCs by UC-MSCs.

We further explored the mechanism by which UC-MSCs increased CD1c+DCs via FLT3L. We reviewed the database of the gene-annotation portal BioGPS (http://biogps.org/#goto=genereport&id=2323), and found among mononuclear cells, T cells had the highest expression of FLT3L. Moreover, the expression of FLT3L on PBMCs was significantly higher than that on UC-MSCs (Fig. 4g). Therefore, there was a possibility that UC-MSCs indirectly regulated CD1c+DCs via FLT3L in other cells of PBMCs. In order to investigate the contribution of FLT3L on T cells in PBMCs to the increase of CD1c+DCs, the CD4+ and CD8+ T cells were extracted from PBMCs. After co-cultured with UC-MSCs, the frequency of CD1c+DCs in CD4−CD8−PBMCs (T cell-depleted PBMCs) was significantly increased as well (Supplementary Fig. 21), which could basically exclude the possibility that UC-MSCs indirectly regulated CD1c+DCs via T cells. Furthermore, in order to study the contribution of FLT3L on PBMCs to the increase in CD1c+DCs, we co-cultured PBMCs with UC-MSCs (PM group) or autologous PBMCs in which the CD1c+DCs were depleted (PP group). After co-culture for 72 h, the frequency of CD1c+DCs was significantly higher in PM group than that in PP group. Moreover, the neutralizing antibody against FLT3L was added to both PM and PP groups, the up-regulation of CD1c+DCs was abolished in PM group, while there was no significant change of CD1c+DCs in PP

group (Fig. 4h). Hence, we could basically exclude the possibility that UC-MSCs indirectly regulate CD1c+DCs via PBMCs. Although PBMCs expressed higher level of FLT3L than MSCs, the CD1c+DCs could not be boosted by PBMCs compared with MSCs, suggesting a necessary but not sufficient role of FLT3L in promoting CD1c+DCs. Then in order to detect whether MSCs directly regulated CD1c+DCs, we isolated CD1c+DCs from healthy subjects and showed that MSCs dramatically facilitated the proliferation and inhibited the apoptosis of CD1c+DCs, which was similar to effects of FLT3L to FLT3 binding leading to increased DC cell proliferation and the inhibition of apoptosis[32]. In deed, we not only uncovered that FLT3L was expressed on UC-MSCs, but also showed that FLT3 was expressed on CD1c+DCs. Functionally, UC-MSCs that were pretreated with siRNA against FLT3L have reduced their ability to increase CD1c+DCs. Thus, the data indicated a role of FLT3L in MSCs-mediated upregulation of CD1c+DCs.

We have previously reported that IFN-γ, which was produced predominantly by lupus CD8+T cells, was a key factor to enhance immune suppressive function of allogeneic MSCs by induction of IDO[26]. We here extended the function of IFN-γ produced by lupus PBMCs or derived from patient serum in regulating MSCs by showing that it increased the expression of FLT3L in MSCs. This was mediated by JAK/STAT signaling pathway.

In addition, we found the UC-MSCs induced CD1c+DCs in the lupus patients exerted tolerogenic properties by decreasing expression of CD80, CD83, CD86 and HLA-DR, decreasing production of TNFα and keeping production of IL-10. These findings suggested the UC-MSCs induced CD1c+DCs could ameliorate immune dysfunction and maintain immune homeostasis, which might be contribute to the benefits of treatment for SLE.

Although our findings revealed that after UC-MSCs treatment, not merely the number of CD1c+DCs increased significantly, more than the anti-inflammatory function of these tolerogenic DCs tended to be stronger than before, the connection between the changes of CD1c+DCs and the benefit of treatment in MSC treated patients was still unknown. To answer this question, we further classified the last 10 SLE patients receiving U-MSCT (patient 12–21) into remission group and no remission group. Compared with no remission group, the increases of both frequency and absolute number of CD1c+DCs (post U-MSCT—pre U-MSCT) were significantly higher in remission group (Supplementary Fig. 6). To sum up, we suggested that the increase of CD1c+DCs number induced by MSCs had contribution to the potential benefit of treatment in MSCs treated SLE patients.

It was reported that MSCs employed various mechanisms to regulate immune cells including DCs. For example, HGF and IL-6 have been reported to affect DC maturation and function[38,39]. We found that in the steady state, MSCs expressed higher levels of IL-6 and HGF compared to FLT3L. Under IFN-γ treatment,

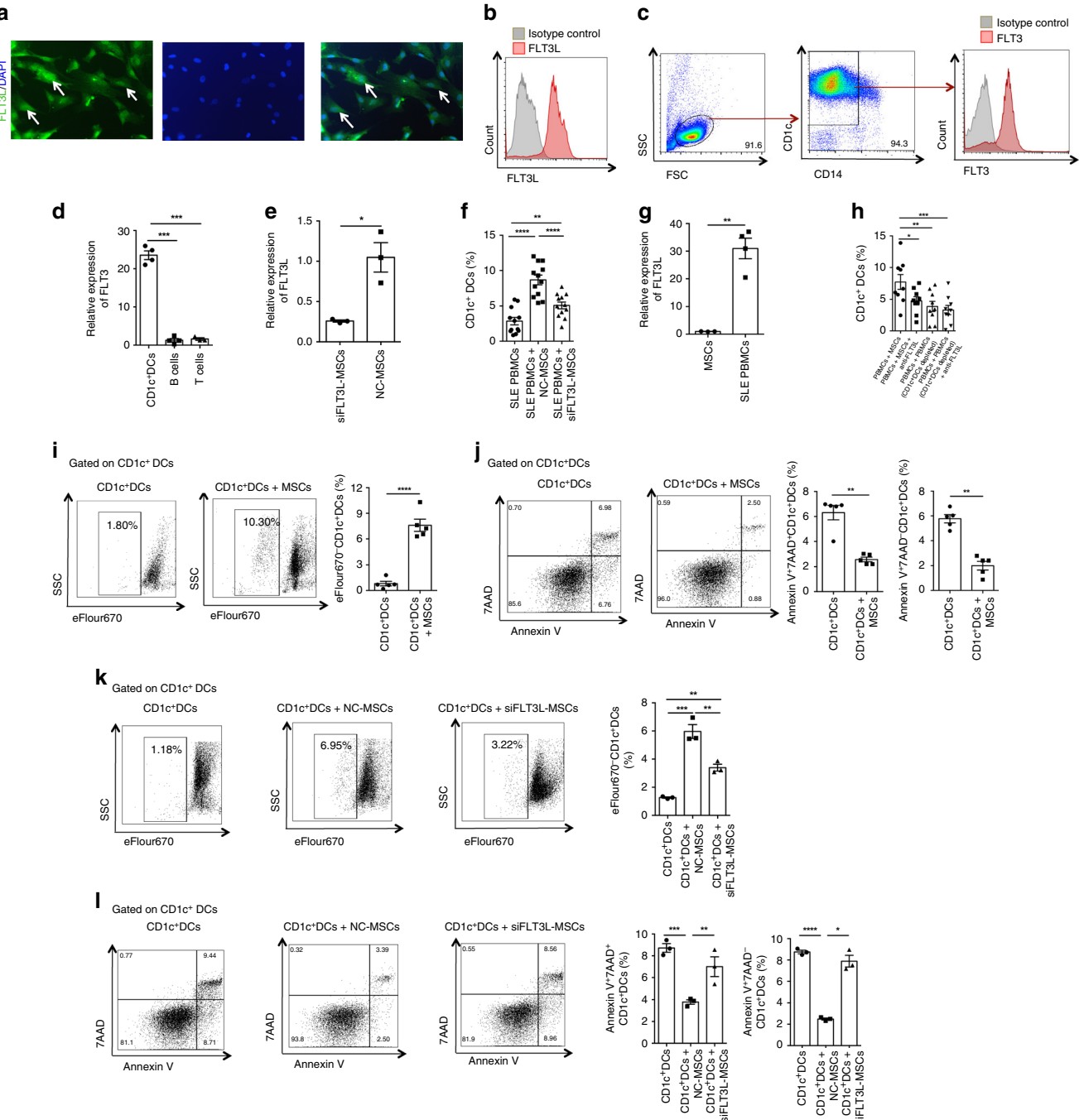

**Fig. 4** UC-MSCs up-regulate CD1c+DCs via FLT3L. **a** The expression of FLT3L on UC-MSCs was assessed by immunofluorescence staining (Green: FLT3L; Blue: DAPI), white arrows represented FLT3L expression. **b** The expression of FLT3L on UC-MSCs was assessed by flow cytometry. **c** CD1c+DCs are isolated and identified by flow cytometry. Flow cytometry analysis showed the expression of FLT3 on CD1c+DCs. **d** The relative expression of FLT3 was compared among isolated CD1c+DCs ($n = 4$), CD19+B cells ($n = 4$) and CD4+T cells ($n = 3$) by real-time PCR. **e** FLT3L siRNA was used to down-regulate FLT3L expression on UC-MSCs (siFLT3L-MSCs), negative control siRNA treated MSCs (NC-MSCs) were served as controls. The inhibition efficiency of FLT3L siRNA was assessed in UC-MSCs by real-time PCR ($n = 3$). **f** FLT3L siRNA was used to down-regulate FLT3L expression on UC-MSCs, then co-cultured with SLE PBMCs, and the frequency of Lin−HLA-DR+CD11c+CD1c+DCs was analyzed ($n = 12$). **g** The expression of FLT3L on SLE PBMCs ($n = 4$) was compared with that on UC-MSCs ($n = 3$) by real-time PCR. **h** SLE PBMCs were co-cultured with UC-MSCs (PM group) or autologous PBMCs in which the CD1c+DCs were deleted (PP group) for 72 h. The frequency of CD1c+DCs was compared between the two groups. The neutralizing antibody against FLT3L was added to both PM and PP groups, the frequency of CD1c+DCs was evaluated in each group ($n = 9$). **i** The proliferative rate of isolated CD1c+DCs cultured alone or co-cultured with UC-MSCs for 72 h ($n = 5$). **j** The apoptotic rate of isolated CD1c+DCs cultured alone or co-cultured with UC-MSCs for 24 h ($n = 5$). **k, l** FLT3L siRNA was used to down-regulate FLT3L expression on UC-MSCs, NC-MSCs were served as controls, then co-cultured with isolated CD1c+DCs, the proliferation (**k**) and apoptosis (**l**) of isolated CD1c+DCs were evaluated ($n = 3$). The experiments were repeated ≥3 times (*$p < 0.05$, **$p < 0.01$, ***$p < 0.001$, ****$p < 0.0001$ by t-test or one-way ANOVA), SEM. Source data were provided as a Source Data file

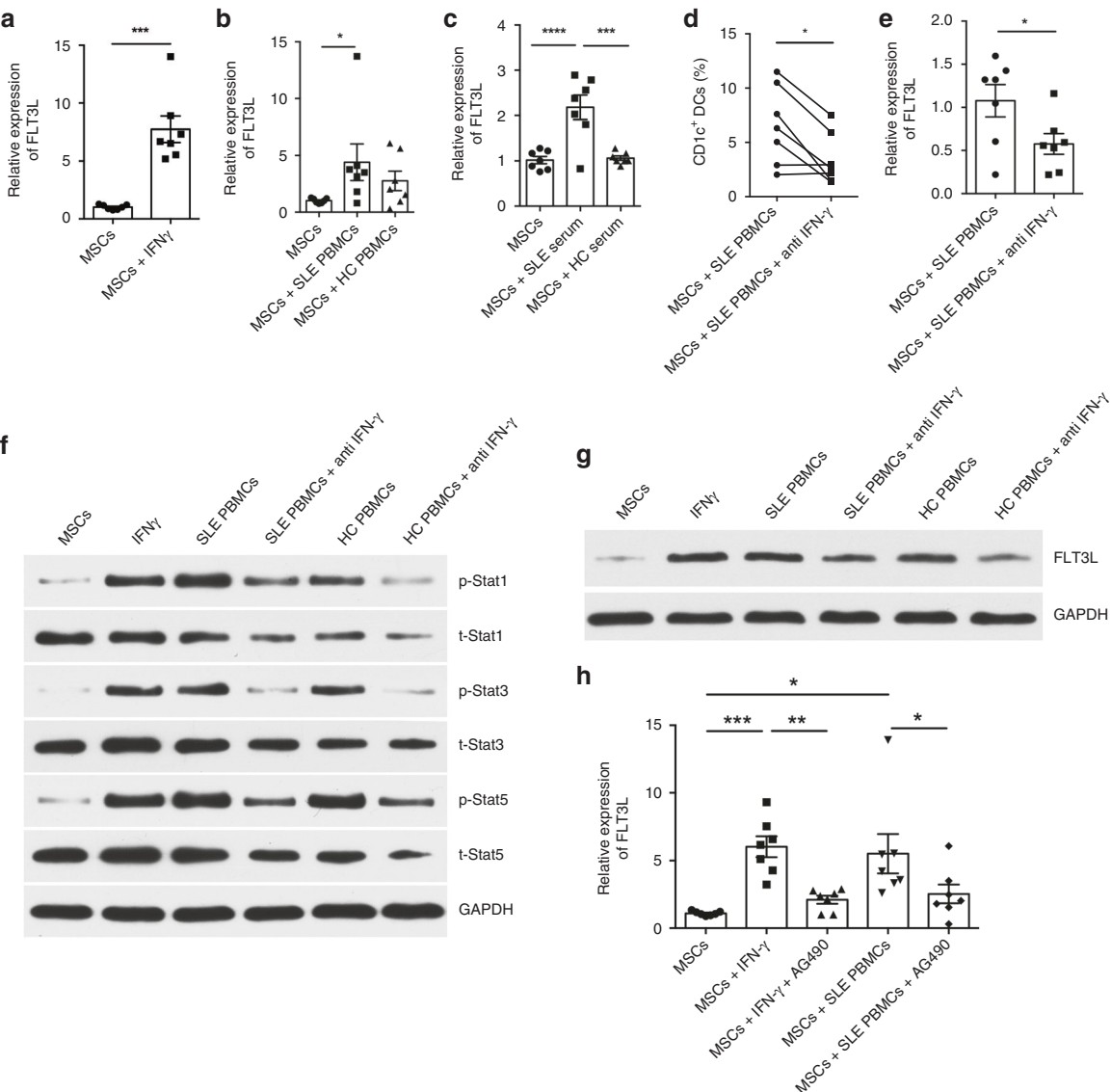

**Fig. 5** IFN-γ promotes FLT3L in UC-MSCs via JAK/STAT pathway. **a**, **b**, **c** The expression of FLT3L in UC-MSCs was analyzed by real-time PCR after stimulated by IFN-γ (20 ng/ml) (**a**), SLE/HC PBMCs (**b**) and SLE/HC serum (**c**) ($n = 7$). **d**, **e** SLE PBMCs were co-cultured with UC-MSCs with or without anti-IFN-γ monoclonal antibody for 72 h. The percentage of CD1c+DCs in PBMCs was analyzed by flow cytometry ($n = 7$) (**d**), and the expression of FLT3L in UC-MSCs was assessed by real-time PCR ($n = 7$) (**e**). **f** Stat1, Stat3, Stat5 and their phosphorylated forms were assessed by western blot analysis after treatment with UC-MSCs alone, UC-MSCs with PBMCs from SLE patients or HC, and UC-MSCs with PBMCs and anti-IFN-γ monoclonal antibody, or recombinant human IFN-γ. **g** FLT3L protein level produced by UC-MSCs was assessed by western blot analysis after treatment with UC-MSCs alone, UC-MSCs with PBMCs from SLE patients or HC, and UC-MSCs with PBMCs and anti-IFN-γ monoclonal antibody, or recombinant human IFN-γ. **h** UC-MSCs were cultured alone, or co-cultured with SLE PBMCs with or without AG490 (20 nM), or treated with IFN-γ with or without AG490. The expression of FLT3L in UC-MSCs was assessed by real-time PCR ($n = 7$). The experiments were repeated ≥3 times (*$p < 0.05$, **$p < 0.01$, ***$p < 0.001$, ****$p < 0.0001$ by $t$-test or one-way ANOVA), SEM. Source data were provided as a Source Data file

MSCs expressed higher levels of IL-6 compared to FLT3L (Supplementary Fig. 17). Blocking either IL-6 (Supplementary Fig. 18) or HGF (Supplementary Fig. 19) had no effect on the progress that MSCs significantly up-regulated the number of CD1c+DCs in vitro (Supplementary Fig. 20). Therefore, we might basically rule out the contribution of IL-6 and HGF on MSCs to promote the up-regulation of CD1c+DCs.

Dendritic cells can be divided into pDCs, CD141+DCs and CD1c+DCs, all with different functions. Therefore, we further investigated the changes of pDCs and CD141+DCs after MSCs treatment. No significant changes of absolute numbers and percentages of pDCs (Supplementary Fig. 7, 12) or CD141+DCs (Supplementary Fig. 8, 13) were observed after MSCs treatment

both in vivo and in vitro. These results might further indicate that, rather than other DC subsets, the function of MSCs treatment on the alteration of DCs in lupus patients mainly focused on CD1c+DC subtype; therefore, combining the previous results shown in the manuscript, there seemed to be more evidence to prove the causative connection between MSCs treatment and the improvement of CD1c+DCs.

For the 145 SLE patients included in this study who donated blood for the cell culture and flow cytometry experiments, one patient was newly onset SLE without any medication, the other 144 patients underwent steroid therapy, and 96 underwent immunosuppressant therapy including cyclophosphamide, mycophenolate mofetil, cyclosporine, azathioprine, leflunomide,

tacrolimus, etc. The potential influence of these medications on tolerogenic DCs should be taken into consideration. First, we excluded the possibility that the decrease of CD1c+DCs in SLE patients was associated with medications. It is reported that glucocorticoids have been widely used to promote tolerogenic DCs development[40,41]. Also, studies based on the generation of tolerogenic DCs have used glucocorticoids to develop new immunotherapies for SLE patients[42]. In addition, several immunosuppressive drugs, such as cyclosporine A, tacrolimus, mycophenolate mofetil, azathioprine, and leflunomide[43–47], have been shown to affect DCs function and result in the inability of DCs to establish effective immune responses, thereby strengthening the tolerogenic circuit. Although the medications taken by patients could increase tolerogenic DCs, the tolerogenic CD1c+DCs of SLE were observed to be lower than that of controls in this study, which further confirmed that the level of CD1c+DCs were significantly decreased in SLE. Second, the contribution of MSCs to the increase of CD1c+DCs should be compared with that of medications. As CD1c+DC significantly increased 24 h after U-MSCT, and the medication kept the same during this period, we could basically ruled out the contribution of the medication to the increase of CD1c+DCs. Furthermore, 10 SLE patients with lupus nephritis were included, who achieved partial remission after immunosuppressive medications treatment for 1 month. No significant changes of absolute numbers and percentages of CD1c+DCs were observed after the treatment (Supplementary Fig. 9), indicating that the mechanism of traditional immunosuppression drugs in successfully treating lupus nephritis might not be associated with the number change of CD1c+DCs. This finding was slightly different from previous literature reporting that the immunosuppressive drugs could increase tolerogenic DCs[43–47]. The contradictory results may be due to several factors including: (1) Most previous studies were in vitro experiment; (2) No previous studies have focused on the changes of CD1c+ DCs.

In summary, we uncovered that CD1c+DCs as critical orchestrators of an immunoregulatory cell network in patients with SLE. We have revealed a previously unrecognized IFNγ-FLT3L-FLT3 axis in allogeneic UC-MSCs-mediated increase in CD1c+DCs in lupus patients. The findings should have implications for MSC therapeutic applications in SLE.

## Methods

**Subjects**. Totally, 166 SLE patients and 78 healthy subjects were included in this study. The average age of the SLE patients and healthy controls were $35.42 \pm 12.19$ and $34.01 \pm 15.18$, respectively, and the gender ratio (female/male) of the SLE patients and healthy controls were 143/23 and 66/12, respectively. The two groups were age- and gender- matched. All the subjects were given informed consents for the collection of peripheral blood samples. The SLE patients were consecutively enrolled according to the following inclusion criteria: All patients fulfilled the 1997 revised criteria of the American College of Rheumatology for SLE[48]. Renal involvement of SLE patients was clinically diagnosed through persistent abnormal urine tests[49]. Partial remission of LN was defined as a value for urinary protein excretion that was between 0.3 and 2.9 g per 24 h, with a serum albumin concentration of at least 3.0 g per deciliter[50]. From January 2016 to January 2019, besides one patient receiving no medications, 145 among 166 SLE patients received conventional therapies including glucocorticoid with/without immunosuppressive treatment, etc. Their demographic characteristics and medications were listed on Supplementary Table 2, and each figure represented which patients have been addressed in Supplementary Table 3. Twenty-one SLE patients refractory to conventional therapies were enrolled in an U-MSCT trial after signing informed consent. All enrolled patients had an SLEDAI score of more than or equal to 8 or had at least one British Isles Lupus Assessment Group (BILAG) grade A or at least two BILAG grade B manifestations. Refractory to treatment was defined as lack of response to conventional immunosuppressive drugs (cyclophosphamide 500–750 mg/m²/month, mycophenolate mofetil ≥1000 mg/day, leflunomide 20 mg/day, azathioprine 100 mg/day, tacrolimus ≥2 mg/day, alone or in combination for more than 6 months) or continued daily doses of at least 20 mg of prednisone or its equivalent. Patients were excluded from the study if they had the following conditions: (1) uncontrolled infection, such as infection, including pneumonia (bacterial, virus, or fungal), pulmonary tuberculosis, hepatitis B and C, skin infection, central nervous system infection; (2) severe organ dysfunction such as heart failure

New York Heart Association functional classification III or IV, hepatic failure, renal failure, or respiratory failure; (3) woman who was pregnant or lactating, or a woman or man who intended to initiate a pregnancy in the following 6 months[51].

Clinical study of UC-MSCs transplantation for lupus patients was registered in Clinical Trial.gov (identifier: NCT01741857). This study was approved by the Ethics Committee at the Drum Tower Hospital of Nanjing University Medical School and was conducted in accordance with the principle set forth under the 1989 Declaration of Helsinki.

**Antibodies and reagents**. The following antibodies (to humans) were used in this study: fluorescein isothiocyanate (FITC)-conjugated anti-human hematopoietic lineage cocktail (eBioscience), phycoerythrin (PE)-conjugated anti-human CD1c (eBioscience), anti-human CD135 (FLT3) (eBioscience), anti-human CD123 (eBioscience), anti-human CD141 (BD Biosciences), allophycocyanin (APC)-conjugated anti-human CD11c (eBioscience), PE-Cy7-conjugated anti-human HLA-DR (eBioscience), BV711-conjugated anti-human CD40 (BioLegend), PE-Dazzle594-conjugated anti-human CD80 (BioLegend), anti-human TNFα (BioLegend), APC-Cy7-conjugated anti-human CD83 (BioLegend), anti-human CD1c (BioLegend), BV605-conjugated anti-human CD86 (BioLegend), BV421-conjugated anti-human IL-10 (BioLegend), PerCP-Cy5.5-conjugated anti-human CD11c (BioLegend) and their respective isotype-matched control antibodies (mouse IgG1κ, mouse IgG2a, mouse IgG2b, Rat IgG1κ) were from eBioscience. Cell Proliferation Dye eFluor™ 670 was from eBioscience. Annexin V/7AAD apoptosis detection kit Annexin V was from BD Biosciences. Anti-FLT3 ligand antibody (ab52648) and its isotype-control Rabbit IgG, monoclonal were both from Abcam. Anti-Flt3 ligand monoclonal antibody (ab9688) was from Abcam, anti-IFN-γ monoclonal antibody was from R&D Systems. Recombinant human FLT3L was from PeproTech. Human FLT3L Quantikine ELISA Kit was from R&D Systems. Human CD1c (BDCA-1)+ DC Isolation Kit was from Miltenyi Biotec.

**Isolation, culture and infusion of UC-MSCs**. Fresh umbilical cords were obtained with informed consents from healthy mothers in local maternity hospitals after normal deliveries. UC-MSCs for each recipient were from one donor. However, since the clinical effect is clear and consistent, we did not specifically state MSCs heterogeneity in this paper. UC-MSCs were prepared by the Stem Cell Center of Jiangsu Province (Beike Bio-Technology). The umbilical cords were rinsed in PBS with added penicillin and streptomycin, the cord blood being removed during this process. The washed cords were cut into 1-mm²-sized pieces and floated in DMEM-LG containing 10% FBS. The pieces of cord were subsequently incubated at 37 °C in humid air with 5% $CO_2$. Nonadherent cells were removed by washing. The medium was replaced every 3 days after the initial plating. When well-developed colonies of fibroblast-like cells appeared after 10 days, the cultures were trypsinized and passaged into a new flask for further expansion. At 80–85% confluence, the adherent cells were detached by treatment with 0.125% trypsin and 0.1% EDTA.

For preservation, UC-MSCs were resuspended in a cryoprotectant solution composed of 90% FBS and 10% dimethyl sulfoxide (DMSO) (Edwards Lifesciences) and stored at −80 °C overnight, then transferred to the vapor phase of a liquid nitrogen tank. Frozen cells were thawed for clinical-scale expansion using the same protocol as described for primary expansion. Cells from passage 2 to 5 were harvested during the second expansion period. After rinsing twice with Plasmalyte-A (Baxter), cell pellets were resuspended in Plasmalyte-A containing 1% or 5% human serum albumin (Hualan Biological Engineering Inc.). The majority of remaining cells were packaged in 25 ml disposable plastic blood bags (10 million cells in 10 ml volume) (Shandong Weigao Group Medical Polymer Company Limited).

Criteria for release of UC-MSCs for clinical use included spindle-shaped morphology, absence of visible clumps and absence of cell supernatant contamination by pathogens, as well as by virus for hepatitis B surface antigen, hepatitis B core antibody, hepatitis C virus antibody, human immunodeficiency virus antibodies I and II, cytomegalovirus IgM, and syphilis antibody, and cell viability greater than 92%. Immunophenotype analysis indicated that the cultured UC-MSCs had positive expressions of CD73, CD105, CD90, and CD29 (>90%) and negative expressions of CD45, CD34, CD14, CD79, and HLA-DR (<2%).

Twenty-one patients underwent UC-MSCs transplantation. All patients underwent U-MSCT and one million cells per kilogram of body weight were administered by intravenous infusion, without adding steroid or other immunosuppressive drugs. After U-MSCT, the doses of steroids as well as immunosuppressive drugs were tapered according to the amelioration of disease conditions. If the clinical index was not improved or if disease activity had not declined, the drug dose was not tapered or new drugs might be chosen. When relapse occurred, the dose of prednisone could be added or new drugs would be given.

**Isolation and culture of PBMCs**. PBMCs were isolated from active lupus patients and healthy controls at the same time point by density gradient centrifugation on Ficoll, and PBMCs were resuspended in PBS containing 1% bovine serum albumin and 0.1% sodium azide. In vitro experiments, recombinant human FLT3L was added to PBMCs. After 72 h culture, cells were harvested for examining by flow cytometry. Also, PBMCs were co-cultured with UC-MSCs for 72 h at a ratio of 10:1, and supernatants were collected after centrifugation.

**Table 3 Primers for real-time polymerase chain reaction**

| Gene | Forward | Reverse |
|------|---------|---------|
| FLT3L | 5′-TCGCTTCGTCCAGACCAAC-3′ | 5′-CCAGGTCAGTGCTCCACAAG-3′ |
| FLT3 | 5′-AGGGACAGTGTACGAAGCTG-3′ | 5′-GCTGTGCTTAAAGACCCAGAG-3′ |
| IL-6 | 5′-ACTCACCTCTTCAGAACGAATTG-3′ | 5′-CCATCTTTGGAAGGTTCAGGTTG-3′ |
| HGF | 5′-GCTATCGGGGTAAAGACCTACA-3′ | 5′-CGTAGCGTACCTCTGGATTGC-3′ |
| GAPDH | 5′-ATGGGGAAGGTGAAGGTCG-3′ | 5′-GGGGTCATTGATGGCAACAATA-3′ |

**Isolation of CD1c⁺DCs.** CD1c⁺DCs were purified with magnetic cell sorting (MACS) using anti-CD1c (BDCA-1) micro beads according to the manufacturer's instructions. In brief, CD19⁺ cells were depleted using anti-CD19-coated magnetic beads, and then CD1c⁺ cells were isolated using biotinylated anti-CD1c and anti-biotin−beads.

**Flow cytometry analysis and ELISA.** For the detection of intracellular cytokines, the PBMCs were stimulated at 37 °C for 6 h with 1 μg/ml LPS (Sigma) or 4 μg/ml CD40L (Invitrogen), in the presence of 10 μg/ml brefeldin A (Enzo), 20 ng/ml phorbol-12-myristate-13-acetate (Enzo) and 1 μg/ml ionomycin (Enzo) during the last 4 h of stimulation. For the detection of changes in surface antigen expression, the PBMCs were stimulated at 37 °C for 6 h with 1 μg/ml LPS (Sigma).

PBMCs were resuspended in PBS containing 1% bovine serum albumin and 0.1% sodium azide. For the staining of surface antigens of cells, they were incubated with FITC-, PE-, APC-, PE-Cy7-, BV711-, PE-Dazzle594-, APC-Cy7-, BV605- or PerCP-Cy5.5-conjugated monoclonal antibodies or their isotype-control antibodies as indicated for 15 min on ice. For the staining of intracellular cytokines, they were incubated with PE-Dazzle594-, or BV421-conjugated monoclonal antibodies or their isotype-control antibodies as indicated for 30 min on ice after permeabilization and fixation. For proliferation assays, CD1c⁺DCs ($1 \times 10^5$/well) were cultured alone or co-cultured with UC-MSCs at a ratio of 10:1, Cell Proliferation Dye eFluor™ 670 was added to the cultures. The incorporation of eFluor™ 670 in CD1c⁺DCs was tested by flow cytometry after 72 h co-culture. For the detection of apoptosis, CD1c⁺DCs ($1 \times 10^5$/well) was cultured alone or with UC-MSCs at a ratio of 10:1. Three days later, CD1c⁺DCs were collected, resuspended in Annexin V binding buffer, and stained with FITC-Annexin V and 7AAD. Annexin V⁺ 7AAD⁺ cells were detected by flow cytometry. We detected the amounts of FLT3L in the culture supernatant solution and/or human serum with ELISA Kits according to the manufacturer's instructions.

**Real-time PCR.** Total RNA from cells isolated using TRIzol reagent (Invitrogen) was reverse-transcribed into complimentary DNA (cDNA) by the use of the PrimeScript RT Master Mix kit (Takara). For quantitative PCR analysis, reactions containing the SYBR Premix EX Taq (Takara), cDNA, and gene primers were run on the StepOnePlus Real-Time PCR Systems and analyzed with StepOne Software V2.1 (Applied Biosystems). Gene primers are listed in Table 3. The relative gene quantification was done by using the $2^{-\triangle\triangle Ct}$ method following normalization to glyceral-dehyde-3-phosphate dehydrogenase (GAPDH).

**Western blot analysis.** Proteins were extracted in RIPA buffer supplemented with EDTA-free protease inhibitor cocktail (Roche Diagnostics) and phosphatase inhibitor cocktail 3 (Sigma). Proteins were run on 10% gradient gel (BioRad) and blotted onto nitrocellulose membrane. Antibodies for immunoblotting were as follows: p-Stat1 (monoclonal rabbit, Abcam), Stat1 (monoclonal rabbit, Abcam), p-Stat3 (monoclonal rabbit, Abcam), Stat3 (monoclonal rabbit, Abcam), p-Stat5 (monoclonal rabbit, Abcam), Stat5 (monoclonal rabbit, Abcam), FLT3L (monoclonal rabbit, Abcam), GAPDH (monoclonal rabbit, CST). Images were captured and analyzed on Tanon−5200 Chemiluminescent Imaging System. Total density of each protein band was determined with ImageJ software. For each sample, the ratio of target protein to GAPDH total density was calculated. All uncropped scans of western blots are presented in Source Data file.

**Small interfering RNA silencing.** The expression of FLT3L, IL-6, and HGF by UC-MSCs was down-regulated by RNA interference technique. FLT3L, IL-6, and HGF small interfering RNA (siRNA) and negative control (non-target control) were synthesized by Biomics Biotechnologies Company Limited. FLT3L siRNA sequence: CUUCAAGAUUACCCAGUCA-UGACUGGGUAAUCUUGAAG, IL-6 siRNA sequence: GGCAAAGAAUCUAGAUGCA-UGCAUCUAGAUUCUUUGCC, HGF siRNA sequence: GUAAAGGACGCAGCUACAA-UUGUAGCUGCGUCCUUUAC, negative control sequence: UUCUCCGAACGUGUCACGU-ACGUGACACGUUCGGAGAA. The siRNAs were transfected into cells with Lipofectamine 2000 (Invitrogen) according to the instruction and the interference effect was examined by real-time PCR.

**Immunofluorescence.** Cells were fixed with 4% paraformaldehyde (PFA) in PBS for 20–30 min and permeabilized in 0.3% Triton X-100 (Sigma) for 10 min.

Nonspecific binding sites were blocked with 10% goat serum for 30 min at room temperature. Cells were then incubated overnight at 4 °C or 1 h at room temperature with primary antibody (Anti-FLT3 ligand antibody). After three rinses in PBS, cells were exposed to goat anti-rabbit immunoglobulin G conjugated to fluorescein isothiocyanate, Alexa-488 (dilution 1:1000) (MultiSciences) for 1 h at room temperature, followed by nuclear staining with DAPI for 5 min. After three rinses in PBS, coverslips were mounted on slides. The cells on coverslips were examined using a biological navigator (FSX100, Olympus).

**Statistical analysis.** We used the t-test or chi-square test for statistical analysis for parametric data and the Mann–Whitney U-test for non-parametric data. One-way analysis of variance was used when there were more than two groups, and then followed by Least-Significant Difference (LSD) test among different groups. We performed statistical analyses with SPSS16.0 software and GraphPad Prism 4.3 and considered a P value <0.05 as significant difference. Data are shown as means ± standard error of mean (SEM).

**Reporting summary.** Further information on research design is available in the Nature Research Reporting Summary linked to this article.

## Data availability

The data that support the findings of this study are available on request from the corresponding author. The source data underlying Figs. 1c–i, 2a–e, 3b–d, 3f–x, 4d–l, 5a–e, 5h, and Supplementary Figs. 1–15 and 17–24 are provided as a Source Data file.

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

## Acknowledgements

This work was funded by grants from the Major International (Regional) Joint Research Project of China (81720108020), Jiangsu Province Major Research and Development Program (BE2015602), Jiangsu Province 333 Talent Grant (BRA2016001). National Natural Science Foundation of China (81601414), Jiangsu Province Six Talent Project (2016-WSN-156), and the Nanjing Outstanding Youth Foundation (JQX17004). Wanjun Chen is supported by the Intramural Research Program of NIDCR, NIH.

## Author contributions

Xinran Yuan, Xiaodong Qin, and Dandan Wang designed the research, performed research, analyzed data, wrote and edited paper. Zhuoya Zhang assisted to perform research, revised the paper. Xiaojun Tang assisted to perform research. Xiang Gao assisted to perform research. Wanjun Chen designed the research, and assisted in the writing and revising of the manuscript. Lingyun Sun provided foundation and finance, designed the research, and assisted in the writing and revising of the manuscript.

## Additional information

**Competing Interests:** The authors declare no competing interests.

