## [Peer Review File · Nature Communications]

Reviewers' comments:

Reviewer #1 (Remarks to the Author):

In this manuscript, the authors demonstrated an interesting correlation between CD1c+ DCs, FLT3L and SLE. It was found that MSCs derived FLT3L can increase CD1c+ DCs by enhancing proliferation and inhibiting apoptosis. This seems to be a potentially new mechanism underlying MSCs mediated immune-regulation. However, the following questions should be carefully addressed.

1. MSCs employ various of mechanisms to regulate immune cells including DCs. For example, HGF (PMID:25114100) and IL6 (PMID: 17510220) have been reported to affect DC maturation and function. Although the authors argued that FLT3L plays a role in regulating DC proliferation and apoptosis, the exact expression levels of FLT3L is unknown as only immunofluorescence and flow cytometry results are shown. Thus, the expression levels of RNA and protein should be carefully examined and the production of FLT3 should be evaluated. The expression of FLT3L should be compared to IL6 and HGF, both in the steady state and under inflammatory cytokines treatment conditions. Block IL6 and HGF and then check the effect on CD1c+ in vitro should be carried out to rule out their contribution. The relative expression and contribution of FLT3L in MSCs should be compared with that of PBMCs.

2. Beside the increased percentage/number of CD1c+ dendritic cells, its function should also be included in the manuscript upon MSC treatment. The function of these cells should be tested using functional markers or assays.

3. FLT3 is a master regulator of the commitment of hematopoiesis to the myeloid cell lineages. In the manuscript, the authors demonstrated that MSCs did not affect CD1c- cells differentiation into CD1c+ DCs using PBMCs. However, the authors should be aware that the PBMC fraction contains only limited DC precursors as compared to bone marrow progenitor cells. Thus it is premature to state "MSCs did not affect CD1c+ DC differentiation."

4. Line 186, the evidence showing that T cells can regulate CD1c+ DCs should be cited.

5. Figure 4b and 4c, in the histogram, "FLT3L" and "Isotype" might be mislabeled.

6. Figure 4c shows FLT3L expression instead of FLT3 expression as described in figure legend and main text.

7. For the apoptosis assay, PI should be used to exclude necrotic cells.

Reviewer #2 (Remarks to the Author):

The manuscript presents a large body of data in support of a series of novel observations in patients with SLE, including some therapeutic results. Therefore the data should be examined carefully.

The data are convincing that levels of CD1c tolerogenic cells are low in SLE patients and that they show negative correlations with several manifestations of the disease (Fig. 1). The changes in FLT3 and its ligand are much smaller (Fig. 2). The text should acknowledge this instead of simply referring to them as significant.

The data on patients who received an infusion of UC-MSCs (Fig. 3) present a serious problem. There is essentially no information as to how the cells were expanded, to what extent, how they were preserved, and how characterized. How many donors of UC were used? Was there a difference in efficacy of different preparations as others have seen with different preparations of MSCs (see McLeod CM, Mauck RL. *Eur Cell Mater.* 2017 Oct 27; 34: 217-231). Where are the controls for a clinical trial for a disease in which the placebo effects can approach 50% (see J, Bass D, Fox NL, Roth D, Gordon D. *Arthritis Rheumatol.* 2017 May; 69(5): 1016-1027)? Clearly the data fall short of current standards. The data on mechanisms (Figs. 4 and 5) are more straight forward.

The paper is in general well written but I think most readers would be grateful if some the sentences were shortened. See lines 45 to 50 and 72 to 76.

In summary, the paper contains some interesting information that is well documented but some that is not.

Reviewer #3 (Remarks to the Author):

This manuscript describes 11 patients treated for refractory lupus nephritis with mesenchymal stem cells. This is an uncontrolled clinical study but the benefits appear striking as previously reported. The authors suggest that a mechanism for the therapeutic effect is via induction of tolerogenic dendritic cells. Flow cytometry studies of peripheral blood from lupus patients shows a lower percentage of CD1c dendritic cells than in healthy controls and in vitro studies show that mesenchymal stem cells expressing Flt3L induce the differentiation of CD1c dendritic cells. The authors suggest that IFN gamma present in the sera of active SLE patients can induce MSCs to upregulate Flt3L which in turn may help to expand CD1c DCs. However a causative connection between the altered percentages of CD1c DCs in MSC treated patients and the benefit of treatment is not shown in this manuscript. The authors have previously reported other mechanisms of immune modulation in MSC treated patients so it is still not clear why the treatment is beneficial.

Dendritic cells can be divided into pDCs, CD141 DCs and CD1c DCs (cDC1 and cDC2), all with different functions. Flt3L is known to expand CD1c DCs and in vitro the DCs that are generated tend to have an immature phenotype that may be tolerogenic. However in vivo the function of these cells is influenced by their environment and they can be matured into immunogenic cells by other stimuli such as CD40 ligation and TLR activation, both of which are likely to occur in lupus patients. It is therefore imperative that the function of the MSC induced DCs in the lupus patients be shown ie their ability to present antigen, activation status (CD80/CD86 expression) cytokine production etc in order to understand whether these are indeed tolerogenic and contributing to potential benefits of the treatment.

There are also several major technical and clinical design problems with the manuscript that make the data very difficult to interpret. First the flow cytometry identification of CD1c DCs shown in Figure 1 is not optimal as some of the DR positive cells are not included. The authors are referred to <https://www.ncbi.nlm.nih.gov/pmc/articles/PMC4673437/figure/fig1/> for typing strategy. Second only percentages of cells are shown but not numbers – in fact the authors use percentage and number interchangeably which may not be correct depending on what has happened to other PBMC subsets. Unless numbers are shown as well as percentages in each figure it is not possible to interpret the data. What happened to the other DC subtypes? Third, many patients were used with (most likely) different patients in each figure. There is no description of how control SLE patients were chosen for each experiment (ie were they consecutively enrolled) or what medications they were taking. A full table of all 100 patients with a list of their demographic characteristics and medications and which figure represents each patient would be needed to address this. In addition it is not clear whether the healthy controls were matched for age and gender. Since the patients before and after treatment were on different amounts of immunosuppressive treatment it is not clear whether the increase in CD1c DCs after treatment just reflects less immunosuppression or whether it reflects less disease activity. It

is not shown whether patients successfully treated just with immunosuppression for their nephritis would manifest the same phenotypic changes.

The in vitro studies are of some interest since they suggest that MSCs can express Flt3L upon stimulation with interferon gamma and that this is induced by SLE cells and serum.

Other comments

Data in Fig 2D is not significant

Figure 3J and 3L not needed

Figure 3P: Flt3L does not appear to have any effect on CD1c DC maturation in vitro. This is contrary to the proposed hypothesis and not consistent with the inhibition experiments shown in the next figure.

Please explain.

Figure 4A and B: are Flt3L and isotype control reversed?

Figure 5F and G need quantitation data graphed and statistics

Dear Dr. Bondar:

Thank you very much for your letter regarding the decision on our paper (NCOMMS-18-26759A). We appreciate the time and efforts that the editor and reviewers devoted to our paper. We have addressed all the insightful and constructive questions and comments raised by the reviewers with additional experimental data, and revised the manuscript accordingly. The changes to the manuscript have been highlighted in yellow color and are also underlined. A *point-by-point* response to the reviewers' questions is provided below.

Reviewer #1 (Remarks to the Author):

In this manuscript, the authors demonstrated an interesting correlation between CD1c⁺ DCs, FLT3L and SLE. It was found that MSCs derived FLT3L can increase CD1c⁺ DCs by enhancing proliferation and inhibiting apoptosis. This seems to be a potentially new mechanism underlying MSCs mediated immune-regulation. However, the following questions should be carefully addressed.

We thank the reviewer for his/her positive comments on our paper and we have addressed all of the questions as outlined below.

1. MSCs employ various of mechanisms to regulate immune cells including DCs. For example, HGF (PMID:25114100) and IL-6 (PMID: 17510220) have been reported to affect DC maturation and function. Although the authors argued that FLT3L plays a role in regulating DC proliferation and apoptosis, the exact expression levels of FLT3L is unknown as only immunofluorescence and flow cytometry results are shown. Thus, the expression levels of RNA and protein should be carefully examined and the production of FLT3 should be evaluated. The expression of FLT3L should be compared to IL-6 and HGF, both in the steady state and under inflammatory cytokines treatment conditions. Block IL-6 and HGF and then check the effect on CD1c⁺ in vitro should be carried out to rule out their contribution. The relative expression and contribution of FLT3L in MSCs should be compared with that of PBMCs.

The reviewer's questions and constructive suggestions are valid and well taken. We might have not described clearly in our previous version that we used multiple methods to examine the expression levels of FLT3L on MSCs, including real-time PCR for mRNA, immunofluorescence, flow cytometry and western blotting for protein as shown in Fig. 4a, b and Fig. 5a, b, c, e, g. According to the reviewer's suggestion, the expression of FLT3L was then compared to IL-6 and HGF on MSCs. We found that in the steady state, MSCs expressed higher levels of IL-6 and HGF compared to FLT3L. Under IFN- γ treatment, MSCs also expressed higher level of IL-6 compared to FLT3L (Supplementary Fig. 17). However, blocking either IL-6 or HGF (Supplementary Fig. 18, 19) had no effects on the upregulation of CD1c⁺DCs number induced by MSCs *in vitro* (Supplementary Fig. 20). Therefore, the data could rule out the contribution of IL-6 and HGF on MSCs to promote the up-regulation of CD1c⁺DCs. Please see the revised manuscript from lines 215 to 219 on page 13, and lines 354 to 363 on page 19 to 20.

On the other hand, we found that the expression of FLT3L on PBMCs was higher than that on MSCs (Supplementary Fig. 22). To study the contribution of FLT3L on PBMCs to the increase in CD1c⁺DCs, we co-cultured PBMCs with MSCs (PM group) or with autologous PBMCs in which the CD1c⁺DCs were depleted (PP group). The frequency of CD1c⁺DCs was significantly higher in PM group than that in PP group after 72h of co-cultures. Moreover, the neutralizing antibody against FLT3L was added to both PM and PP groups, the up-regulation of CD1c⁺DCs was abolished in PM group, while there was no significant change of CD1c⁺DCs in PP group (Supplementary Fig. 23). Hence, we could basically conclude that the FLT3L in MSCs had much more contribution to the increase of CD1c⁺DCs than that in PBMCs. Please see the revised manuscript from lines 226 to 230 on page 13, and lines 314 to 322 on page 18.

In addition, the expression of FLT3 in CD1c⁺DCs was further evaluated by real-time PCR. Compared with CD4⁺T cells and CD19⁺B cells, CD1c⁺DCs expressed higher levels of FLT3, which was consistent with our previous flow cytometry results (Fig. 4d). Please see the revised manuscript from lines 210 to 212 on page 13.

2. Beside the increased percentage/number of CD1c⁺ dendritic cells, its function should also be included in the manuscript upon MSC treatment. The function of these cells should be tested using functional markers or assays.

To address this question, we performed more experiments to study the influence of MSCs to the function of CD1c⁺DCs by their functional markers (CD40/CD80/CD83/CD86 expression), HLA-DR expression and cytokines production (IL-10/TNF α). We found that, *in vitro*, LPS-mediated upregulation of CD83, CD86 and HLA-DR expressions in CD1c⁺DCs were significantly decreased in SLE PBMCs co-cultured with MSCs than without MSCs (Fig. 3u, v). Similarly, LPS-induced TNF α produced by CD1c⁺DCs was also significantly decreased in SLE PBMCs co-cultured with MSCs than without MSCs (Fig. 3w), whereas there was no significant difference of IL-10 production between these two groups (Fig. 3x).

In vivo, we investigated the effect of MSCs transplantation on the function of CD1c⁺DCs by the same markers and cytokine production as studied *in vitro* experiment. We found that after MSCs transplantation, the expression of the co-stimulatory molecules (CD40/CD80/CD83/CD86) and HLA-DR of CD1c⁺DCs in freshly isolated peripheral blood of treated SLE did not change significantly. However, the CD80, CD86 and HLA-DR expression of CD1c⁺DCs in SLE PBMCs that were stimulated with LPS significantly decreased after MSCs transplantation compared to before transplantation (Fig. 3q, r). Similarly, the levels of TNF α of CD1c⁺DCs in SLE PBMCs were also significantly decreased after MSCs transplantation (Fig. 3s), while the levels of IL-10 showed no changes after MSCs infusion (Fig. 3t). Collectively, our data demonstrate that in lupus microenvironment, MSCs not only increase the number of CD1c⁺DCs, but also tend to drive CD1c⁺DCs towards a more tolerogenic phenotype. Please see the revised manuscript from lines 168 to 195 on page 9 to 10.

3. FLT3 is a master regulator of the commitment of hematopoiesis to the myeloid cell lineages. In the manuscript, the authors demonstrated that MSCs did not affect CD1c⁺ cells differentiation into CD1c⁺ DCs using PBMCs. However, the authors should be aware that the PBMC fraction contains only limited DC precursors as compared to

bone marrow progenitor cells. Thus it is premature to state “MSCs did not affect CD1c⁺ DC differentiation.”

We thank the reviewer for this comments and apologize for our pre-mature conclusion in our previous version of the manuscript. We have now revised our statement to “UC-MSCs could not drive CD1c⁻ PBMCs to differentiate into CD1c⁺ DCs”. Please see the revised manuscript from lines 235 to 238 on page 13 to 14.

4. Line 186, the evidence showing that T cells can regulate CD1c⁺ DCs should be cited.

We thank the reviewer for this comment and apologize for any confusion the statement might have caused. We didn't describe it properly in our previous version of the manuscript. We did not mean there was literature reporting that T cells could regulate CD1c⁺DCs. In our own experiments, we found there was a trend that UC-MSCs could up-regulate CD1c⁺DCs in CD4⁻CD8⁻PBMCs (T cell-depleted PBMCs). Then we enrolled three more subjects in this revision, and found UC-MSCs significantly up-regulated CD1c⁺DCs in CD4⁻CD8⁻PBMCs (n=8), which could basically concluded that UC-MSCs up-regulating CD1c⁺DCs was T cells-independent (Supplementary Fig. 21). Please see the revised manuscript from lines 221 to 226 on page 13, lines 302 to 313 on page 18.

5. Figure 4b and 4c, in the histogram, “FLT3L” and “Isotype” might be mislabeled. This mislabeling has been corrected in the revised manuscript. Please see the revised manuscript in Fig. 4b and 4c.

6. Figure 4c shows FLT3L expression instead of FLT3 expression as described in figure legend and main text.

We apologize for the mislabeling and any confusion it might have caused. This mistake has been corrected in the revised manuscript. Please see the revised manuscript in Fig. 4c.

7. For the apoptosis assay, PI should be used to exclude necrotic cells.

We thank the reviewer for this suggestion and agreed necrotic cells in the apoptosis assay should be excluded. Since we used phycoerythrin (PE)-conjugated anti-human CD1c to gate CD1c⁺DCs, we included 7AAD panel to exclude necrotic cells. We found that the percentage of necrotic cells (Annexin V⁻7AAD⁺) was nearly undetectable (less than 1%). In addition, the percentages of the late apoptotic cells and/or dead cells (Annexin V⁺7AAD⁺CD1c⁺DCs) and early apoptotic cells (Annexin V⁺7AAD⁻CD1c⁺DCs) were both significantly decreased after CD1c⁺DCs were co-cultured with UC-MSCs for 24h. However, the inhibition of FLT3L in UC-MSCs significantly abrogated the capacity of UC-MSCs in suppressing the apoptosis of CD1c⁺DCs (Fig. 4h, j). The findings suggested a key role of FLT3L in MSCs-mediated decrease of CD1c⁺DCs apoptosis. Please see the revised manuscript in Fig. 4h and 4j on page 14.

Reviewer #2 (Remarks to the Author):

The manuscript presents a large body of data in support of a series of novel observations in patients with SLE, including some therapeutic results. Therefore the data should be examined carefully.

The data are convincing that levels of CD1c tolerogenic cells are low in SLE patients and that they show negative correlations with several manifestations of the disease (Fig. 1). The changes in FLT3 and its ligand are much smaller (Fig. 2). The text should acknowledge this instead of simply referring to them as significant.

The data on patients who received an infusion of UC-MSCs (Fig. 3) present a serious problem. There is essentially no information as to how the cells were expanded, to what extent, how they were preserved, and how characterized. How many donors of UC were used? Was there a difference in efficacy of different preparations as others have seen with different preparations of MSCs (see McLeod CM, Mauck RL. *Eur Cell Mater.* 2017 Oct 27;34:217-231). Where are the controls for a clinical trial for a disease in which the placebo effects can approach 50% (see J, Bass D, Fox NL, Roth D, Gordon D. *Arthritis Rheumatol.* 2017 May;69(5):1016-1027)? Clearly the data fall short of current standards.

The data on mechanisms (Figs. 4 and 5) are more straight forward.

The paper is in general well written but I think most readers would be grateful if some the sentences were shortened. See lines 45 to 50 and 72 to 76.

In summary, the paper contains some interesting information that is well documented but some that is not.

1. We thank the reviewer's positive comments on our paper. We acknowledged that although both FLT3L and CD1c⁺DCs were significantly decreased, and FLT3 was significantly increased in SLE patients, the changes of FLT3 and its ligand were smaller than that of CD1c⁺DCs in SLE patients. Please see the revised manuscript from lines 107 to 109 on page 6.

2. We added more detailed information about how the cells were prepared and characterized. Please see the revised manuscript on page 23 to 24: lines 471 to 484 (how UC-MSCs were prepared), lines 485 to 494 (how UC-MSCs were preserved, and how they were expanded), lines 489 to 490 (to what extent), and lines 495 to 502 (how UC-MSCs were characterized). Umbilical cords were obtained from healthy mothers in local maternity hospitals after normal deliveries. UC-MSCs for each recipient were from one donor (lines 471 to 473).

3. We understand that MSCs heterogeneity might exist as the reviewer stated. The immunoregulatory function of MSCs from different sources might be different. However, our animal model data revealed that MSCs from human bone marrow and umbilical cord showed the comparable effect in treating arthritis model (unpublished data in revision). Moreover, in clinical settings, we have compared the therapeutic effect between human bone marrow and umbilical cord derived MSCs in treating drug-refractory lupus patients, and both sources of MSCs efficiently ameliorated disease activity without significant difference (Cell Transplant 2013, 22:2267-77; Stem Cell Rep 2018, 10:933-41). Since the clinical effect is clear and consistent, we did not specifically state MSCs heterogeneity in this paper. However, we added a statement in our revised manuscript (lines 473 to 474 on page 23) to reflect this point of the reviewer and will pay more attention on this question in our future studies.

4. We agree with the reviewer that we do not have control group in the present study. All enrolled patients suffered severe conventional drug-resistant disease without other optional therapies. It is unethical to include control group with these patients. One

multi-center, randomized and controlled clinical trial (NCT03580291) has been approved and is in progress, which will compare U-MSCT and mycophenolate mofetil (MMF) in inducing disease remission in active lupus nephritis patients.

5. Several sentences have been shortened and improved to respond to the reviewer's comments. Please see the revised manuscript from lines 40 to 45 on page 2, lines 67 to 71 on page 3.

Reviewer #3 (Remarks to the Author):

1. This manuscript describes 11 patients treated for refractory lupus nephritis with mesenchymal stem cells. This is an uncontrolled clinical study but the benefits appear striking as previously reported. The authors suggest that a mechanism for the therapeutic effect is via induction of tolerogenic dendritic cells. Flow cytometry studies of peripheral blood from lupus patients shows a lower percentage of CD1c dendritic cells than in healthy controls and in vitro studies show that mesenchymal stem cells expressing Flt3L induce the differentiation of CD1c dendritic cells. The authors suggest that IFN gamma present in the sera of active SLE patients can induce MSCs to upregulate Flt3L which in turn may help to expand CD1c DCs. However a causative connection between the altered percentages of CD1c DCs in MSC treated patients and the benefit of treatment is not shown in this manuscript. The authors have previously reported other mechanisms of immune modulation in MSC treated patients so it is still not clear why the treatment is beneficial.

We thank the reviewer for his/her insightful comments. It has been reported before (including our studies) that MSCs treatment could regulate several immune cells such as T cells, B cells, NK cells and macrophages etc, which all might play a part in the improvement of disease activity of SLE. Our study here has revealed that MSCs treatment significantly up-regulated the number of another type of immune cells, CD1c⁺DCs (Fig. 3a, b, c, e, f, g; Supplementary Fig. 4), which was positively accompanied with remarkable amelioration of the disease activity (Fig. 3i; Supplementary Fig. 5). Moreover, the frequency of CD1c⁺DCs was significantly correlated to SLEDAI scores (Fig. 1e). Hence, we concluded that the altered

percentages of CD1c⁺DCs in MSCs treated patients were associated with the benefit of treatment.

Furthermore, we have added 10 more patients in the revised manuscript to respond to the reviewer's comment and further strengthen our conclusions (Supplementary Table. 1). The new 10 SLE patients receiving U-MSCT were divided into remission group and no remission group according to the SLEDAI score at the 1-month follow-up. Six patients showed remission with SLEDAI scores < 8 at 1-month follow-up, and four patients showed no remission with SLEDAI scores \geq 8 at 1-month follow-up. Compared with no remission group, the increases in both frequency and absolute number of CD1c⁺DCs (post U-MSCT – pre U-MSCT) were significantly higher in the remission group (Supplementary Fig. 6). The results further support our conclusion that the increase in CD1c⁺DCs in MSCs treated patients is associated with the benefit of treatment. Please see the revised manuscript from lines 145 to 146 on page 8, lines 343 to 353 on page 19.

2. Dendritic cells can be divided into pDCs, CD141 DCs and CD1c DCs (cDC1 and cDC2), all with different functions. Flt3L is known to expand CD1c DCs and in vitro the DCs that are generated tend to have an immature phenotype that may be tolerogenic. However in vivo the function of these cells is influenced by their environment and they can be matured into immunogenic cells by other stimuli such as CD40 ligation and TLR activation, both of which are likely to occur in lupus patients. It is therefore imperative that the function of the MSC induced DCs in the lupus patients be shown ie their ability to present antigen, activation status (CD80/CD86 expression) cytokine production etc in order to understand whether these are indeed tolerogenic and contributing to potential benefits of the treatment.

We agree with the reviewer that this is an important point and have thus performed more experiments to confirm that MSCs induced CD1c⁺DCs in lupus inflammatory environment are indeed tolerogenic. We collected blood samples before and 24h after U-MSCT from 10 refractory SLE patients and assessed their functional alterations namely their activation status, antigen presenting ability (HLA-DR expression) and cytokine production.

(1) Activation status

First we examined the co-stimulatory (CD40/CD80/CD83/CD86) molecules expressed by CD1c⁺DCs from SLE patients before and after U-MSCT. We found that, LPS-mediated upregulation of CD80 and CD86 expressions in CD1c⁺DCs were significantly decreased in SLE patients after MSCs transplantation compared to before treatment (Fig. 3q), indicating that MSCs treatment make CD1c⁺DCs more tolerogenic.

(2) HLA-DR expression

LPS-mediated upregulation of HLA-DR expression in CD1c⁺DCs was also significantly decreased after MSCs transplantation (Fig. 3r), suggesting that MSCs treatment might reduce the antigen-presenting ability of CD1c⁺DCs *in vivo*.

(3) Cytokine production

We also determined the changes of cytokine production of CD1c⁺DCs in the SLE patients before and after U-MSCT. We found that upon LPS or CD40L stimulation, the level of TNF α of CD1c⁺DCs was significantly decreased in SLE patients after MSCs transplantation (Fig. 3s, Supplementary Fig. 14), while the level of IL-10 showed no significant change in the MSCs treated patients (Fig. 3t, Supplementary Fig. 15), suggesting that MSCs treatment make CD1c⁺DCs more tolerogenic.

In addition, we studied the functional change of CD1c⁺DCs in SLE PBMCs after co-cultured with UC-MSCs. Consistent to the above *in vivo* data, LPS-mediated up-regulation of CD83, CD86 and HLA-DR expression as well as TNF α production in CD1c⁺DCs were significantly decreased when co-cultured with UC-MSCs than without UC-MSCs (Fig. 3u, v, w), whereas there was no significant difference of IL-10 production between the two groups (Fig. 3x). Please see the revised manuscript from lines 168 to 195 on page 9-10, lines 337 to 342 on page 19.

3. There are also several major technical and clinical design problems with the manuscript that make the data very difficult to interpret. First the flow cytometry identification of CD1c DCs shown in Figure 1 is not optimal as some of the DR positive cells are not included. The authors are referred to <https://www.ncbi.nlm.nih.gov/pmc/articles/PMC4673437/figure/fig1/> for typing strategy.

We thank the reviewer for the comments. Our previous gating strategy was referred to a previous publication (<http://www.bloodjournal.org/content/bloodjournal/122/6/932/F1.large.jpg?width=800&height=600&carousel=1>). In response to the reviewer's suggestion, we have revised the gating strategy to include all HLA-DR positive cells (Fig. 1a). Using the new gating strategy, we re-analyzed our previous data concerned and revised related figures. We found the new gating strategy didn't change our conclusions. Please see the revised manuscript in Fig. 1; 2e; 3a, b, c, e, f, g, m, n, p; 4f; 5d; Supplementary Fig. 1, 3, 10, 21, 24.

4. Second only percentages of cells are shown but not numbers – in fact the authors use percentage and number interchangeably which may not be correct depending on what has happened to other PBMC subsets. Unless numbers are shown as well as percentages in each figure it is not possible to interpret the data. What happened to the other DC subtypes?

We appreciate the reviewer's suggestions. We have re-calculated the absolute numbers and percentages of the CD1c⁺DCs according to the new gating strategy in Fig. 1c and 1d (Comparison of CD1c⁺DCs between SLE and healthy controls), Fig. 3b (Comparison of CD1c⁺DCs between Pre U-MSCT and 24h Post U-MSCT), Fig. 3f (Comparison of CD1c⁺DCs among Pre U-MSCT, 24h Post U-MSCT and 72h Post U-MSCT). We now show that (1) Both the absolute numbers and percentages of the CD1c⁺DCs was significantly lower in SLE patients than healthy controls (Fig. 1c, d); (2) Both the absolute numbers and percentages of the CD1c⁺DCs significantly increased 24h after U-MSCT (Fig. 3b, c); (3) Both the absolute numbers and percentages of the CD1c⁺DCs were significantly increased 72h after U-MSCT (Fig. 3f, g). Nevertheless, we co-cultured purified CD1c⁺DCs with MSCs, and found MSCs remarkably promoted the proliferation and suppressed the apoptosis of CD1c⁺DCs (Fig. 4g, h), which further confirmed the conclusion that MSCs treatment could increase the number of CD1c⁺DCs. Please see the revised manuscript from lines 86 to 88 on page 4, lines 130 to 131 on page 8.

Moreover, in this revision, 10 more SLE patients receiving U-MSCT were included, and the absolute numbers and percentages of the CD1c⁺DCs were evaluated before

and after U-MSCT. Consistently, both the absolute numbers and percentages of the CD1c⁺DCs increased significantly after U-MSCT (Supplementary Fig. 4), which confirmed our previous results. Please see the revised manuscript from lines 130 to 131 on page 8.

Moreover, we repeated the *in vitro* experiment: (1) PBMCs from 8 SLE patients were co-cultured with MSCs. After co-culture, both the absolute numbers and percentages of the CD1c⁺DCs in SLE PBMCs significantly increased (Supplementary Fig. 11), which was consistent with our previous *in vitro* results; (2) PBMCs from 5 SLE patients were cultured with FLT3L treatment at the concentration of 10ng/ml. After the FLT3L treatment, both the absolute numbers and percentages of the CD1c⁺DCs in SLE PBMCs were significantly increased (Supplementary Fig. 2), which was consistent with our previous *in vitro* data. Please see the revised manuscript from line 161 to 163 on page 9, lines 112 to 114 on page 6.

As mentioned by the reviewer, dendritic cells could be divided into pDCs, CD141⁺DCs and CD1c⁺ DCs, therefore, we further investigated the changes of pDCs and CD141⁺DCs after MSCs treatment *in vivo* and *in vitro*: (1) No significant changes of absolute numbers and percentages of pDCs were observed after MSCs treatment both *in vivo* and *in vitro* (Supplementary Fig. 7, 12). (2) No significant changes of absolute numbers and percentages of CD141⁺DCs were observed after MSCs treatment both *in vivo* and *in vitro* (Supplementary Fig. 8, 13). These results further indicate that, rather than other DC subsets, the function of MSCs treatment on the alteration of DCs in lupus patients mainly focused on CD1c⁺DC subtype. Please see the revised manuscript from lines 146 to 149 on page 8, lines 165 to 167 on page 9, lines 364 to 373 on page 20.

5. Third, many patients were used with (most likely) different patients in each figure. There is no description of how control SLE patients were chosen for each experiment (ie were they consecutively enrolled) or what medications they were taking. A full table of all 100 patients with a list of their demographic characteristics and medications and which figure represents each patient would be needed to address this. In addition it is not clear whether the healthy controls were matched for age and

gender. Since the patients before and after treatment were on different amounts of immunosuppressive treatment it is not clear whether the increase in CD1c DCs after treatment just reflects less immunosuppression or whether it reflects less disease activity. It is not shown whether patients successfully treated just with immunosuppression for their nephritis would manifest the same phenotypic changes.

These are valid questions and constructive suggestions, and they are well taken. The SLE patients were consecutively enrolled according to the following inclusion criteria: All patients fulfilled the 1997 revised criteria of the American College of Rheumatology for SLE (Hochberg, M.C. Updating the American College of Rheumatology revised criteria for the classification of systemic lupus erythematosus. *Arthritis and rheumatism* 40, 1725 (1997)). Renal involvement of SLE patients was clinically diagnosed through persistent abnormal urine tests (Hahn, B.H., et al. American College of Rheumatology guidelines for screening, treatment, and management of lupus nephritis. *Arthritis care & research* 64, 797-808 (2012)). The demographic characteristics and medications of all the 166 patients were listed in the (Table 1, 2; Supplementary Table 1, 2). Each figure represented which patients have been addressed in Supplementary Table 3. In addition, The average age of the SLE patients and healthy controls were 35.42 ± 12.19 and 34.01 ± 15.18 , respectively, and the gender ratio (female/male) of the SLE patients and healthy controls were 143/23 and 66/12, respectively. The two groups were age- and gender- matched. Please see the revised manuscript from line 414 to 429 on page 21 to 22, Supplementary Table 1, 2, 3 on page 30-36.

Ten SLE patients with lupus nephritis were included, who achieved partial remission after immunosuppressive medications treatment for 1 month. No significant changes of absolute numbers and percentages of CD1c⁺DCs were observed after the treatment (Supplementary Fig. 9), indicating that the mechanism of traditional immunosuppression drugs in successfully treating lupus nephritis might not be associated with CD1c⁺ DCs. Please see the revised manuscript from lines 149 to 151 on page 8, lines 391 to 404 on page 21.

6、Data in Fig 2D is not significant

We thank the reviewer's for point. We detected the relationship between the two molecules, and found there was a trend towards that FLT3L was negatively correlated with FLT3, but with no statistical significance. The results might be due to the relative small sample size and heterogeneity of SLE patients. We have added a statement to reflect this point of the reviewer. Please see the revised manuscript from lines 106 to 107 on page 6.

7、 Figure 3J and 3L not needed

In response to the reviewer's comment, we have deleted Fig. 3j and 3l.

8、 Figure 3P: Flt3L does not appear to have any effect on CD1c DC maturation in vitro. This is contrary to the proposed hypothesis and not consistent with the inhibition experiments shown in the next figure. Please explain.

In Fig. 3p, there has been a trend that FLT3L might increase the percentage of CD1c⁺DCs, but with no statistical significance. This might be due to the heterogeneity of SLE patients and relative small sample size. In line of this notion, we added 5 more SLE patients, whose PBMCs were cultured with FLT3L treatment. We combined these new data with the original data and found that, besides MSCs treatment, FLT3L treatment could also significantly increased the percentage of CD1c⁺ DCs (Fig. 3p). Please see the revised manuscript from lines 164 to 165 on page 9.

9、 Figure 4A and B: are Flt3L and isotype control reversed?

We apologize for the mislabeling in our previous submission. This mistake has been corrected in the revised manuscript. Please see the revised manuscript in Fig. 4b.

10、 Figure 5F and G need quantitation data graphed and statistics

We thank the reviewer for this comment. The total density of each protein band was determined with Image J software. For each sample, the ratio of target protein to GAPDH total density was calculated. Please see the revised manuscript in Supplementary Fig. 25, 26.

We again thank you for your helpful suggestions. We believe that our manuscript has been significantly improved after we have addressed the comments and concerns. We hope that our manuscript is now ready for publication at Nature Communications.

Best wishes

REVIEWERS' COMMENTS:

Reviewer #1 (Remarks to the Author):

In general, most of my questions have been satisfactorily addressed in the revised manuscript entitled "Mesenchymal stem cells increase CD1c+ tolerogenic dendritic cells by up-regulating FLT3 ligand in human systemic lupus erythematosus",

I only have one suggestion:

The presented experimental data clearly demonstrate that PBMCs alone express ten times more FLT3L than MSCs (Fig. S22), although the authors provided evidence that PBMCs could not boost CD11c+ DCs through PBMC, suggesting a necessary but not sufficient role of FLT3L in promoting CD1c+ DCs. These data (as represented in S22 and S23) should be shown in the main text and discussed further to make it clearly noticed by readers.

Reviewer #2 (Remarks to the Author):

The authors have adequately addressed my criticisms.

Reviewer #3 (Remarks to the Author):

No further comments

Point-by-point response to Reviewers

Reviewer #1 (Remarks to the Author):

The presented experimental data clearly demonstrate that PBMCs alone express ten times more FLT3L than MSCs (Fig. S22), although the authors provided evidence that PBMCs could not boost CD11c+ DCs through PBMC, suggesting a necessary but not sufficient role of FLT3L in promoting CD1c+ DCs. These data (as represented in S22 and S23) should be shown in the main text and discussed further to make it clearly noticed by readers.

We thank the reviewer's constructive suggestions. The Fig. S22 and S23 have been moved to Fig. 4g and 4h, respectively. We further discussed about it in the Discussion.

Reviewer #2 (Remarks to the Author):

The authors have adequately addressed my criticisms.

We thank the reviewer's supports.

Reviewer #3 (Remarks to the Author):

No further comments.

We thank the reviewer's comments.